# Active Learning for Image Segmentation with Binary User Feedback

## Abstract

Deep learning algorithms have depicted commendable performance in a variety of computer vision applications. However, training a robust deep neural network necessitates a large amount of labeled training data, which is time-consuming and labor-intensive to acquire. This problem is even more serious for an application like image segmentation, as the human oracle has to hand-annotate each and every pixel in a given training image, which is extremely laborious. Active learning algorithms automatically identify the salient and exemplar samples from large amounts of unlabeled data, and tremendously reduce human annotation effort in inducing a machine learning model. In this paper, we propose a novel active learning algorithm for image segmentation, with the goal of further reducing the labeling burden on the human oracles. Our framework identifies a batch of informative images, together with a list of semantic classes for each, and the human annotator merely needs to answer whether a given semantic class is present or absent in a given image. To the best of our knowledge, this is the first research effort to develop an active learning framework for image segmentation, which poses only binary (yes/no) queries to the users. We pose the image and class selection as a constrained optimization problem and derive a linear programming relaxation to select a batch of (image-class) pairs, which are maximally informative to the underlying deep neural network. Our extensive empirical studies on three challenging datasets corroborate the potential of our method in substantially reducing human annotation effort for real-world image segmentation applications.

## 1 Introduction

Semantic segmentation (labeling every pixel in an image to the category it belongs to) is one of the core tasks of visual recognition and is extensively used in a variety of applications, including autonomous driving, medical imaging and video surveillance among others (Ghosh et al., 2020). With the advent and popularity of deep learning, several deep architectures have been studied for image segmentation, which have depicted state-of-the-art results (Zhu et al., 2019; Yuan et al., 2020; Liu et al., 2021). However, for these models to work reliably, a large amount of training data (in the form of pixel-level annotated images) is required, which requires significant time and human labor. Thus, an algorithm to reduce human annotation effort is critically important to train deep learning models for image segmentation applications.

*Active Learning (AL)* algorithms identify the most informative samples from vast amounts of unlabeled data (Settles, 2010). This tremendously reduces the human annotation effort in training a machine learning model, as only the samples that are selected by the algorithm need to be labeled manually. Further, since the model gets trained on the exemplar samples from the data, it typically depicts better generalization performance than a passive learner, where the training data is sampled at random. AL has been successfully used in a variety of applications, including computer vision (Yoo & Kweon, 2019), text analysis (Tong & Koller, 2001), bioinformatics (Osmanbeyoglu et al., 2010) and medical diagnosis (Gorriz et al., 2017) among others. The growing popularity of deep learning has motivated research in the field of deep active learning, to efficiently train the data-hungry deep learning models (Ren et al., 2021).

The paucity of human labor and the need to use it more efficiently is even more pronounced for an application like image segmentation, due to the enormous time and effort associated with labeling

every pixel in an image. This necessitates specialized query and annotation mechanisms for the AL algorithms to be feasible in a real-world setting. In this paper, we propose a novel AL algorithm to address this challenging problem, in an effort to alleviate the labeling burden on human oracles [1] while inducing a deep learning model for image segmentation. Our algorithm queries a batch of (image-class) pairs and for each pair, poses the question: "*Does the image $i$ contain the semantic class $j$?*" [2] The human annotator merely has to provide a binary "*yes / no*" feedback for each query. This is depicted in Figure 1. Providing such feedback is extremely easy and less prone to annotation errors; it is also significantly less time-consuming and burdensome than providing pixel-level annotations. Our contributions in this paper can be summarized as follows:

- We present a novel AL framework for image segmentation, which poses only binary ("*yes / no*" ) queries regarding the presence / absence of a semantic class in a given image. To our knowledge, this is the first active learning framework for semantic image segmentation which poses only binary queries to the human annotators.

- We pose the image and class selection as a constrained optimization problem, and derive a linear programming relaxation to select a batch of (image-class) pairs, which are maximally informative to the underlying deep neural network.

- We conduct user studies to estimate the time and human effort required to annotate an image at the pixel-level, region-level and binary-level (our method). This can provide valuable insights and enable us to study the trade-off between the human annotation effort and the generalization capability of the trained deep neural network, for different categories of annotation strategies.

- We conduct extensive empirical studies on three benchmark datasets to study the performance of our framework against competing baselines.

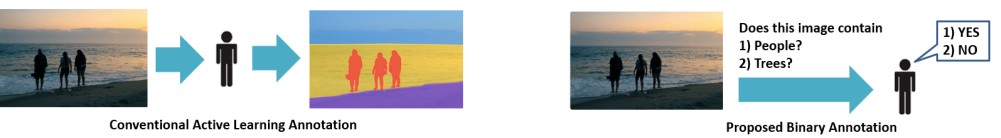

Figure 1: Figure showing the conventional active learning query (left) and the proposed binary query mechanism (right). Best viewed in color.

## 2 RELATED WORK

**Active Learning:** AL is a well-researched problem in the machine learning community (Settles, 2010; Zhan et al., 2022). Uncertainty sampling is the most common strategy for active learning, where unlabeled samples with the highest prediction uncertainties are queried for annotation. Several strategies have been explored to quantify uncertainty, such as Shannon's entropy (Li & Guo, 2013; Joshi et al., 2010), disagreement among a committee of classifiers regarding the label of a sample (Freund et al., 1997), the Fisher information matrix (Hoi et al., 2006), mutual information between the labeled and unlabeled samples (Guo & Greiner, 2007) among others. The growing success and popularity of deep learning have motivated researchers to explore the problem of deep active learning (DAL), where the goal is to select the informative unlabeled samples to train a deep neural network (Ren et al., 2021). Common DAL techniques include incorporating a loss prediction module to predict the loss value of an unlabeled sample and querying samples accordingly (Yoo & Kweon, 2019), selecting informative unlabeled samples for AL and simultaneously, searching for the best neural architectures on-the-fly (Geifman & El-Yaniv, 2019), a sampling technique based on diverse gradient embeddings (*BADGE*) (Ash et al., 2020), a technique which captures the information balance between the uncertainty of underlying softmax probability and the label variable and queries samples accordingly (Woo, 2023) and a technique to select a coreset of samples, such that the model learned over the selected subset is competitive for the remaining data points (Sener & Savarese, 2018). Techniques based on adversarial learning have depicted particularly impressive

---

[1]the terms *user*, *annotator*, *oracle* and *labeler* are used interchangeably in this paper

[2]the term *class* is used to mean *semantic class* in this paper

performance in this context (Sinha et al., 2019; Mayer & Timofte, 2020; Zhang et al., 2020). A segment of AL research has focused on weak / noisy labels, where annotators can provide noisy annotations or can provide annotations at different levels of precision (Olmin et al., 2023; Wu et al., 2017; Younesian et al., 2021; Lu et al., 2017).

Beyond the conventional label query, a body of research in AL has focused on the development of novel query and annotation mechanisms to further reduce the labeling burden on human users. Binary feedback mechanism has been studied, where the active learner queries a pair of images, and the human annotator has to specify whether or not the two images belong to the same category (Joshi et al., 2010; Fu et al., 2014). In another variant, the learner queries an unlabeled image together with a class label, and the human annotator has to specify whether the selected image belongs to that class (Hu et al., 2019; Bhattacharya & Chakraborty, 2019). Along similar lines, AL has been exploited in clustering, where a pair of samples is queried and the oracles need to specify whether or not the samples in a pair correspond to the same cluster (Biswas & Jacobs, 2012). Although the query mechanism is binary, these methods query the label of an image as a whole, and not the presence of a semantic class within an image, and hence are not directly applicable to the problem of image segmentation.

**Active Learning for Image Segmentation:** Providing pixel-level annotations to train an image segmentation model is a time-consuming and expensive process. To address this challenge, weakly supervised semantic segmentation techniques have been developed, such as providing the presence or absence of classes in an image during training (Xu et al., 2014; Pinheiro & Collobert, 2015), pointing to an object of interest (Bearman et al., 2016), bounding box annotations (Papandreou et al., 2015), free-form squiggles (Lin et al., 2016) and noisy web tags (Ahmed et al., 2014). However, these methods utilized the weak supervision only during model training (as a term in the training loss function) and did not use active learning to identify the informative images or the semantic classes within an image.

As in conventional AL, uncertainty and diversity based metrics have been exploited for AL in the context of semantic segmentation (Yang et al., 2017). Metrics like view-point entropy have been studied for multi-view datasets (Siddiqui et al., 2020). Xie *et al.* proposed *DEAL*, a difficulty aware AL algorithm for image segmentation, which focused on the difficulty of different semantic areas in selecting samples for annotation (Xie et al., 2020). A body of research has focused on identifying the informative regions in an image and getting them annotated by the human labelers, rather than the entire image. Various strategies have been explored to identify the informative regions, such as deep reinforcement learning (Casanova et al., 2020), uncertainty quantification using superpixel entropies (Kasarla et al., 2019), informativeness, combined with annotation cost and the spatial coherency of an image (Mackowiak et al., 2018), margin-based sampling combined with diversity (Shin et al., 2021) and self-consistency under equivariant transformations (Golestaneh & Kitani, 2020). Although annotating image regions is less strenuous than providing pixel-level annotations, it still requires the human oracles to meticulously label all the pixels in the queried regions, which can be quite time-consuming, particularly if the queried region involves multiple semantic classes. In contrast, our framework requests only binary feedback regarding the presence / absence of specific classes in an image, which requires much lesser annotation effort and facilitates an easier mode of interaction between the user and the system. We now describe our framework.

## 3 PROPOSED FRAMEWORK

### 3.1 PROBLEM FORMULATION

Consider an active image segmentation problem where we are given a labeled training set $L$ and an unlabeled set $U$. Let $N$ denote the number of unlabeled images, $N = |U|$. Images in $L$ are provided with pixel-level annotations. Let $w$ be the deep neural network trained on $L$, and $C$ be the number of semantic classes in the dataset. We are given a query budget $B$ and a parameter $C_{max}$ which denotes the maximum number of classes that can be queried per image (to ensure that the queries are distributed across a large number of images). Our objective is to select a batch of images, together with a list of classes for each image for binary user query, such that the total number of queries does not exceed the budget $B$, and the user response about the presence/absence of the semantic classes augments maximal information to the deep learning model.

In order to identify the optimal set of images and semantic classes to be queried, we need a metric to quantify the utility score of a batch of (image-class) pairs. We used a criterion based on class presence uncertainty and image redundancy for this purpose. The first criterion ensures that we query those (image-class) pairs where there is maximal uncertainty regarding the presence of the given class in the given image; the redundancy criterion ensures that we query a diverse set of images in our batch and avoid duplicate image queries. These are detailed below.

**Computing Class Presence Uncertainty:** Let $p_{ij}$ denote the probability that image $i$ contains the semantic class $j$ (computed using the current deep neural network $w$, as the average probability of pixels belonging to the semantic class $j$ within image $i$). We used Shannon's entropy to compute the prediction uncertainty of the presence of semantic class $j$ in image $i$:

$$H_{ij} = -p_{ij} \log p_{ij} - (1 - p_{ij}) \log(1 - p_{ij}) \tag{1}$$

Using this, we computed a confidence matrix $G \in \Re^{C \times N}$, where $G(j, i)$ denotes the confidence of the deep model in predicting the presence of class $j$ in image $i$ (high entropy corresponds to low confidence and vice versa):

$$G(j, i) = \frac{\alpha}{H_{ij}} \quad i = 1, \dots N, \ j = 1, \dots C \tag{2}$$

where $\alpha$ is a constant.

**Computing Image Redundancy:** We computed a redundancy matrix $R \in \Re^{N \times N}$, where $R(i, j)$ denotes the redundancy between images $x_i$ and $x_j$ in the unlabeled set. The cosine similarity was used to quantify the redundancy between a pair of samples; negative values were replaced with 0, so that $R$ contains only non-negative entries:

$$R(i, j) = max(0, cos(\mathcal{F}(x_i), \mathcal{F}(x_j))) \tag{3}$$

where $cos(\mathcal{F}(x_i), \mathcal{F}(x_j)) = \frac{\mathcal{F}(x_i)^\top \mathcal{F}(x_j)}{||\mathcal{F}(x_i)|| . ||\mathcal{F}(x_j)||}$ and $\mathcal{F}(x)$ denotes the deep feature representation of image $x$. A low value of $R(i, j)$ implies that images $x_i$ and $x_j$ have low redundancy between them. Cosine similarity has been previously used to compute similarity in AL research, with promising results (Coleman et al., 2022). Depending on the application, other metrics can be used to compute the uncertainty and redundancy terms.

### 3.2 ACTIVE SAMPLING FRAMEWORK

Given $G$ and $R$, our objective is to query a batch of (image-class) pairs such that in each pair, the deep model has low confidence in predicting the presence of the given class in the given image, and the queried images have minimal redundancy among them. We define a binary matrix $M \in \{0, 1\}^{N \times C}$, where each row corresponds to an unlabeled image and each column corresponds to a semantic class. A value of 1 in a row denotes that the image should be selected for annotation, and the position(s) of 1 in a particular row of $M$ denote the semantic class(es) that should be used to pose the binary queries for this image. We also define a binary vector $v \in \{0, 1\}^{N \times 1}$ where $v_i = 1$ denotes that image $x_i$ is selected for annotation, and $v_i = 0$ denotes that it is not selected. The active selection of (image-class) pairs can thus be posed as the following optimization problem:

$$\begin{aligned}
\min_{M, v} \quad & \text{Tr}(MG) + \lambda v^\top R v \\
s.t. \quad & \langle M, E \rangle = B \\
& (M.e)_i \leq C_{max}, \forall i \\
& v_i = \min(1, (M.e)_i), \forall i \\
& v_i, M_{ij} \in \{0, 1\}, \forall i, j
\end{aligned} \tag{4}$$

where $\lambda > 0$ is a weight parameter governing the relative importance of the two terms, $E$ is a matrix of size $N \times C$ (same size as $M$) with all entries 1, $e$ is a vector of size $C \times 1$ with all entries 1, $B$ is the labeling budget, $\langle ., . \rangle$ denotes the inner product operator and Tr denotes the trace of a matrix. The first term in the objective function denotes that the deep model has low confidence in predicting the presence of the selected semantic classes in the corresponding selected images; the second term ensures that the selected images have minimal redundancy among them. The first constraint denotes the total number of queries posed by $M$ is equal to the specified budget; the

second constraint ensures that the number of 1s in each row of $M$ is less than or equal to $C_{max}$, that is, the number of queries posed for each image is less than or equal to the pre-specified limit $C_{max}$; the third constraint denotes that $v_i$ is equal to 1 if there is at least one entry with value 1 in row $i$ of $M$ (image $x_i$ is selected for annotation), and $v_i$ is equal to 0 if all the entries in row $i$ of $M$ have value 0 (image $x_i$ is not selected); the fourth constraint denotes that $v$ is a binary vector and $M$ is a binary matrix. We now present a theorem to solve this optimization problem.

**Theorem 1.** *The optimization problem defined in Equation (4) can be expressed as an equivalent linear programming (LP) problem.*

Please refer to Section A.1 of the Appendix for the proof of this theorem.

We relax the integer constraints into continuous constraints and solve the problem using an off-the-shelf LP solver. After obtaining the continuous solution, we recover the integer solution using a rounding approach where the $B$ highest entries in $M$ are reconstructed as 1 and the other entries as 0, observing the constraints. The pseudo-code of our algorithm is depicted in Algorithm 1 (for one active learning iteration).

---

**Algorithm 1** The Proposed Active Learning Algorithm with Binary User Feedback

---

**Require:** Labeled training set $L$, unlabeled set $U$, query budget $B$, parameters $\alpha, C_{max}$ and $\lambda$, a deep neural network architecture for image segmentation

1: Train the deep model on the training set $L$
2: Compute the confidence matrix $G$ using the probabilities of the trained deep model (Equation (2))
3: Compute the redundancy matrix $R$ (Equation (3))
4: Solve the LP problem in Equation (8) in the Appendix after relaxing the integer constraints
5: Round the solution to derive the matrix $M$
6: Select the unlabeled images and the corresponding semantic classes to pose the binary queries based on the entries in $M$
7: Update the deep model with the user response to the binary queries (detailed in Section F.1 in the Appendix)

---

## 4 EXPERIMENTS AND RESULTS

### 4.1 DATASETS

We used three challenging datasets to study the performance of our framework: ($i$) **Flickr-Landscapes** (Park et al., 2019); ($ii$) **Cityscapes** (Cordts et al., 2016); and ($iii$) **PASCAL VOC12** (Hariharan et al., 2011). All these are benchmark datasets commonly used to validate the performance of image segmentation algorithms.

### 4.2 COMPARISON BASELINES

We used a total of five methods as comparison baselines that annotate images at the pixel-level, region-level and binary-level. These are detailed below.

**Pixel-level annotation**: In this category, a batch of unlabeled images were queried and all the pixels of all the queried images were annotated. We used two AL algorithms to query a batch of unlabeled images: *Entropy* (Settles, 2010), a commonly used AL method which selects samples with the highest degree of uncertainty as computed by entropy (the entropy of an image in our image segmentation application, was computed as the average entropy of every pixel in the image, obtained from the softmax probabilities furnished by the deep network); and *Coreset* (Sener & Savarese, 2018), a widely used AL technique which queries a batch of images such that a model trained on the queried subset is competitive for the remaining data samples.

**Region-level annotation**: Here, a batch of regions were queried from the unlabeled images and all the pixels in the queried regions were annotated. We used the region-based active learning (*RAL*) method proposed by Kasarla *et al.* (Kasarla et al., 2019) where the SLIC algorithm was used to

compute the superpixel of an image, and the regions with the highest uncertainties (defined by the superpixels) were queried for annotation.

**Binary-level annotation**: In this category, binary queries were posed regarding the presence / absence of specific semantic classes in the unlabeled images (similar to our method). This is the first AL framework with binary-level annotation for image segmentation; we hence used the following methods as comparison baselines: *Random-Random (RR)*, which randomly selects a subset of images and randomly queries $B$ semantic classes from the selected images; and *Entropy-Entropy (EE)*, where a batch of images were selected based on the entropy of the underlying model and the semantic classes producing the highest prediction entropy values were queried from each.

We used the *DeepLabV3+* model with the ResNet101 backbone (pre-trained on ImageNet) as our base model due to its promising performance in image segmentation applications (Chen et al., 2018). The same architecture was used for all the baseline methods, for fair comparison.

**Evaluation Metrics:** The *mean intersection-over-union (mIoU)* was used as the evaluation metric, as commonly done in image segmentation research (Chen et al., 2018). Since our comparison baselines span different categories of annotation, we also used the *annotation time* as an evaluation metric.

### 4.3 EXPERIMENTAL SETUP

Each dataset was divided into three parts: $(i)$ an initial training set $L$; $(ii)$ an unlabeled set $U$; and $(iii)$ a test set. The number of images in the initial training, unlabeled and test sets were $1,500$, $1,200$ and $1,000$ respectively for all three datasets. All the images in $L$ were provided with pixel-level annotations. A query budget $B$ (taken as 200 for Cityscapes and PASCAL and 400 for Flickr) was imposed in each AL iteration, and the experiments were conducted for 25 AL iterations. The query budget denotes the number of binary queries that can be posed (for the binary-level annotation methods, *RR*, *EE* and our method) or the number of image regions that can be queried (for the region-level annotation method, *RAL*). However, since we had 1200 images in our unlabeled set, using a query budget of 200 for the pixel-level annotation baselines would have exhausted the unlabeled pool after 6 AL iterations. We hence set the query budget to $48 (= 1200/25)$ in each AL iteration for the pixel-level baselines, so that the unlabeled pool is completely exhausted after 25 AL iterations. Also, since each queried image was annotated at the pixel level for *Entropy* and *Coreset*, these baselines represent an upper bound on the AL performance among the methods studied.

After each AL iteration, the selected samples were annotated and appended to the training set; the deep neural network was retrained and tested on the test set. The objective was to study the improvement in performance on the test set with increasing number of label queries. The value of $\alpha$ in Equation (2) was set as 1, the parameter $C_{max}$ in Equation (4) was taken as 5, and the weight parameter $\lambda$ in Equation (4) was taken as 1 for all the datasets. All the results were averaged over 3 runs (with different training, unlabeled and test sets) to rule out the effects of randomness.

### 4.4 IMPLEMENTATION DETAILS

Please refer to Section F of the Appendix for details on implementation and model parameters. Please refer to Section F.1 of the Appendix for details on updating the deep neural network with binary user feedback. We also provide a few visual illustrations showing the performance of our binary query AL framework (in Section F.2 of the Appendix).

### 4.5 USER STUDY TO ESTIMATE ANNOTATION TIME

To accurately estimate the human annotation time (and hence, effort) required to annotate an image at the pixel-level, region-level and binary-level, we conducted a user study. 10 images were selected at random from each of the three datasets. For each image, the following tasks were posed:

$(i)$ Annotators were asked to segment each image at the pixel level with the different categories of objects and mark each category with a different color (pixel-level annotation)
$(ii)$ Annotators were asked to annotate all the pixels within a given region (super-pixel) of an image with the different categories of objects and mark each category with a different color (region-level annotation)

($iii$) Annotators were asked a question regarding the presence of an object in each image and had to provide a binary response: "YES / NO" (binary-level annotation)

Annotators were provided with the *LabelMe* annotation tool (Russell et al., 2007) to segment the images. The time taken for each annotation task was noted. The annotators were also asked to provide a rating, denoting the ease of annotation for each task, on a scale of 1 to 10, 1 being VERY DIFFICULT and 10 being VERY EASY. Each image was annotated (at the pixel, region, and binary levels) by 3 human annotators independently.

| Annotation Task | Flickr | | Cityscapes | | PASCAL VOC12 | |
|---|---|---|---|---|---|---|
| | Time | Ease | Time | Ease | Time | Ease |
| Pixel-level | 7.8±2.9mins | 5.5±1.2 | 37.5±6.3mins | 3.6±1.6 | 18.2±4.3mins | 5.2±1.7 |
| Region-level | 1.6±1.2mins | 7.3±2.7 | 3.6±0.7mins | 5.5±1.8 | 2.7±1.1mins | 6.7±2.3 |
| Binary-level | 2±0.3secs | 10±0.0 | 4±0.8secs | 10±0.0 | 3±1.4secs | 10±0.0 |

Table 1: User study results. The table reports the average time (and ease of annotation) to annotate **one** complete image at the pixel-level, **one** region within an image at the pixel-level, and to answer **one** binary query posed for a given image, for the three datasets. The results were averaged across all images for a given dataset and all annotators.

The user-study results are reported in Table 1, which depicts the average time (and ease of annotation) across all images and annotators, for the three datasets. The resolution of each image was $513 \times 513$ for Flickr, $768 \times 768$ for Cityscapes and $513 \times 513$ for PASCAL VOC. As evident from the table, pixel-level annotation entailed the maximum amount of time (and hence, human labor). Annotating a given region within an image took considerably less amount of time. As expected, binary-level annotations were the most efficient in terms of time and took only a few seconds for each image. We also note that the pixel-level annotations were the most difficult to provide, followed by region-level annotations. All the annotators reported that binary annotations were the easiest and the most convenient to provide and it consistently received the highest rating of 10. This user study demonstrates the tremendous savings in human annotation effort that can be achieved by the proposed binary-level annotation technique for image segmentation applications. Note that the user study was conducted to estimate the annotation time for the three annotation tasks, which will be used in our empirical analysis (detailed below). To train the DeepLabV3+ model in our experiments, we used the ground truth annotations that are provided for each dataset, since it will be extremely time-consuming to obtain human annotations for all the training images used in our study.

### 4.6 ACTIVE LEARNING PERFORMANCE

The active learning performance results are shown in Figure 2. In each graph, the $x$-axis denotes the iteration number and the $y$-axis denotes the mean IoU on the test set. From the results, we conclude the following:

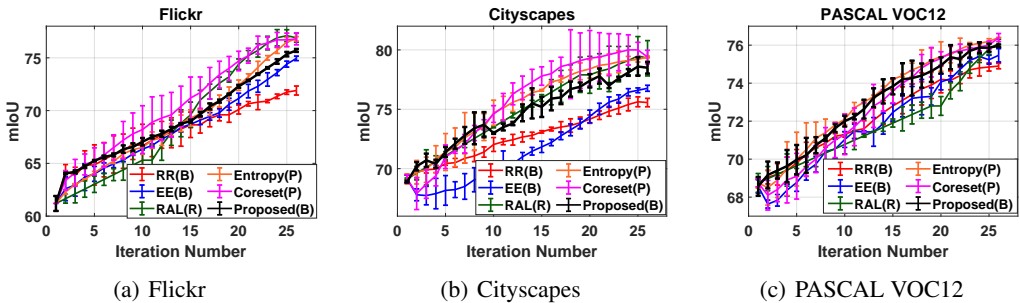

(a) Flickr        (b) Cityscapes        (c) PASCAL VOC12

Figure 2: Active Learning performance comparison. The $x$-axis denotes the iteration number and the $y$-axis denotes the mean IoU on the test set. Query budget = 200 for Cityscapes and PASCAL and 400 for Flickr in each AL iteration. Here, **B** denotes binary-level annotation, **R** denotes region-level annotation and **P** denotes pixel-level annotation. Best viewed in color.
The proposed method comprehensively outperforms the two other AL techniques that utilize binary user feedback: *RR* and *EE*. In almost all the iterations across all three datasets, our framework depicts

| Dataset | RR(B) | EE(B) | RAL(R) | Entropy(P) | Coreset(P) | Proposed(B) |
|---|---|---|---|---|---|---|
| **Flickr** | $71.9 \pm 0.44$ | $74.95 \pm 0.21$ | $76.9 \pm 0.41$ | $76.8 \pm 0.58$ | $76.8 \pm 0.57$ | $75.7 \pm 0.13$ |
| **Cityscapes** | $75.56 \pm 0.37$ | $76.76 \pm 0.25$ | $79.30 \pm 1.48$ | $79.4 \pm 0.07$ | $79.4 \pm 0.52$ | $78.5 \pm 0.47$ |
| **PASCAL** | $74.9 \pm 0.17$ | $75.46 \pm 0.35$ | $76.1 \pm 0.23$ | $76.4 \pm 0.17$ | $76.4 \pm 0.29$ | $75.96 \pm 0.06$ |

Table 2: Final mIoU achieved by all the methods after 25 AL iterations. Here, **B** denotes binary-level annotation, **R** denotes region-level annotation and **P** denotes pixel-level annotation.

| Dataset | Binary-Level | Region-Level | Pixel-Level |
|---|---|---|---|
| **Flickr** | 5.56 | 266.67 | 156 |
| **Cityscapes** | 5.56 | 300 | 750 |
| **PASCAL** | 4.16 | 225 | 364 |

Table 3: Approximate total time (**in hours**) to be expended for annotation (for the binary-level, region-level and pixel-level methods) over 25 AL iterations for all the three datasets. Query budget = 200 for Cityscapes and PASCAL and 400 for Flickr in each AL iteration. Query budget denotes the number of binary queries answered for binary-level annotation methods, and number of image regions annotated for the region-level annotation methods. Pixel-level annotation methods annotate all the $1,200$ unlabeled images at the pixel-level (48 images in each AL iteration for all the datasets).

a better mIoU value compared to these two baselines. The final mIoU achieved by our method after 25 AL iterations is also higher than *RR* and *EE*, for all three datasets. This shows that our algorithm can successfully identify the exemplar (image-class) pairs which augment maximal information to the deep learning model, thereby enabling it to attain much better generalization capabilities.

The *RAL* method (which requires human users to annotate pixels within given image regions) as well as the *Entropy* and *Coreset* methods (which requires users to annotate all the pixels in a given image) marginally outperform the proposed algorithm (for the Flickr and Cityscapes datasets). *Coreset* depicts the best performance for Cityscapes and Flickr while *Entropy* depicts the best performance for PASCAL VOC. Table 2 shows the final mIoU attained by all the methods after 25 AL iterations. We note that *RAL*, *Entropy* and *Coreset* all achieve a marginally higher mIoU than our method. However, these methods also entail a significantly higher human annotation effort than our binary query framework. Table 3 depicts an estimate of the total annotation time (in hours) that has to be expended over the 25 AL iterations, for all the methods studied. These figures were obtained by multiplying the values in Table 1 by the number of annotations performed in each AL iteration and the total number of AL iterations. For instance, for the Cityscapes dataset, the time for pixel-level annotation was computed as: 37.5mins (time taken to annotate one image at the pixel-level) $\times$ 48 (no. of images annotated in each AL iteration) $\times$ 25 (no. of AL iterations); similarly, the time for region-level annotation was computed as: 3.6mins (time taken to annotate the pixels in one region in an image) $\times$ 200 (number of regions annotated in each AL iteration) $\times$ 25 (no. of AL iterations); and the time for the proposed binary annotation was computed as: 4secs (time taken to answer one binary query) $\times$ 200 (number of binary queries answered in each AL iteration) $\times$ 25 (no. of AL iterations). From Figure 2 and Table 3, it is evident that our method requires substantially less annotation time and effort, while producing mIoU values that are comparable to *RAL*, *Entropy* and *Coreset*. For the PASCAL VOC dataset for instance, the final mIoU achieved by our binary query framework is 75.96, and the difference is less than $0.5\%$ compared to the values achieved by *RAL*, *Entropy* and *Coreset* (Table 2). However, the total annotation time required by the region-level (*RAL*) and pixel-level annotation (*Entropy* and *Coreset*) methods are $54.08$ times and $87.5$ times greater than our method respectively (Table 3). These results corroborate the promise and potential of our binary query and annotation technique to substantially reduce human annotation effort, with a marginal loss in performance, in an application like image segmentation, where annotating a single data instance is extremely time-consuming and laborious. From Table 3, we also note that region-level annotation can sometimes take more time than pixel-level annotation, depending on the number of regions annotated and the resolution of the images.

### 4.7 STUDY OF BACKBONE NETWORK ARCHITECTURE

In this experiment, we studied the effect of the backbone network architecture used in the DeepLabV3+ model (we used ResNet-101 as the default backbone architecture). The results on the

Cityscapes dataset (with query budget 200) using the XceptionNet (Chollet, 2017) and ResNet50 backbones are shown in Figure 3. Our framework once again outperforms the binary-level annotation baselines *RR* and *EE* and depicts comparable performance to the region-level (*RAL*) and pixel-level (*Entropy* and *Coreset*) annotation baselines. Table 4 depicts the final mIoU values attained by all the methods after 25 AL iterations; since we have only changed the backbone network architecture (and not the query budget), the total annotation time computed in Table 3 for the Cityscapes dataset is also applicable for this experiment. From Table 4, we note that, for the Xception backbone, our binary query framework depicts the highest mIoU after 25 AL iterations; for the ResNet-50 backbone, our algorithm's final mIoU is marginally less than that of *RAL, Entropy* and *Coreset*. However, as evident from Table 3, the total annotation time required by the region-level (*RAL*) and pixel-level annotation (*Entropy* and *Coreset*) methods are 53.95 times and 134.89 times greater than our method respectively. Our framework thus depicts comparable (and sometimes, marginally better) performance than the region-level and pixel-level annotation baselines, and is significantly more efficient in terms of the total annotation time required for the entire experiment. This shows the robustness of our framework to the backbone network architecture.

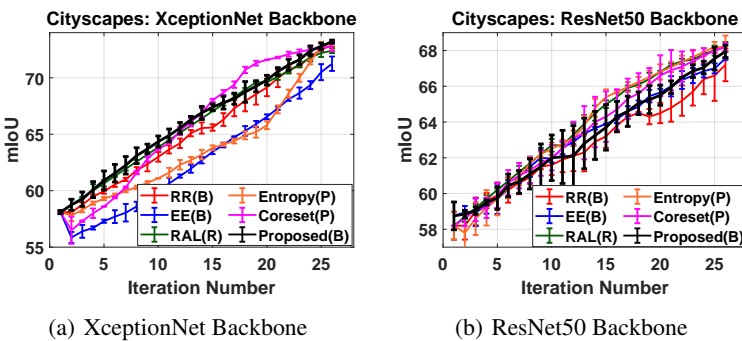

(a) XceptionNet Backbone         (b) ResNet50 Backbone

Figure 3: Study of backbone network architecture on the Cityscapes dataset. Query budget = 200. Best viewed in color.

| Backbone | RR(B) | EE(B) | RAL(R) | Entropy(P) | Coreset(P) | Proposed(B) |
|---|---|---|---|---|---|---|
| **Xception** | $72.9 \pm 0.18$ | $71.25 \pm 0.63$ | $72.4 \pm 0.31$ | $72.8 \pm 0.38$ | $72.8 \pm 0.43$ | $73.2 \pm 0.11$ |
| **ResNet-50** | $67.2 \pm 0.91$ | $67.5 \pm 0.07$ | $68.2 \pm 0.26$ | $68.2 \pm 0.67$ | $68.2 \pm 0.29$ | $67.95 \pm 0.36$ |

Table 4: Final mIoU achieved by all the methods after 25 AL iterations (as shown in Figure 3) are depicted in the table. Here, **B** denotes binary-level annotation, **R** denotes region-level annotation and **P** denotes pixel-level annotation.

*We also conducted the following experiments, which are reported in the Appendix, due to space constraints: study of query budget (Section B); ablation study (Section C); analysis of the computation time of all the methods (Section D); study of the parameter $C_{max}$ (Section E); study of the initial training set size (Section G); and comparison against the fully supervised baseline (Section H).*

## 5   CONCLUSION AND FUTURE WORK

In this paper, we proposed a novel active learning framework for semantic image segmentation, which poses only binary queries regarding the presence / absence of a semantic class in a given image. To the best of our knowledge, this is the first research effort to develop such an active query mechanism in the context of image segmentation. We posed the image and class selection as a constrained optimization problem and derived an LP relaxation to identify a batch of (image-class) pairs for active query. Our empirical results demonstrated the promise and potential of our framework to drastically reduce human annotation effort in training a deep neural network for semantic segmentation applications. We hope this research will motivate the development of novel AL algorithms, particularly for applications where labeling a single data instance involves significant manual work. As part of future research, we plan to explore GPU-based parallel algorithms (such as the one proposed in (Li et al., 2011)) to improve the computational overhead of solving the LP problem.

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

# A APPENDIX

## A.1 PROOF OF THEOREM 1

We provide the proof of Theorem 1 in this section. For better readability, we restate the theorem and the optimization problem in Equation (4):

$$
\begin{aligned}
\min_{M,v} \quad & \mathrm{Tr}(MG) + \lambda v^\top R v \\
s.t. \quad & \langle M, E \rangle = B \\
& (M.e)_i \leq C_{max}, \forall i \\
& v_i = \min(1, (M.e)_i), \forall i \\
& v_i, M_{ij} \in \{0,1\}, \forall i,j
\end{aligned}
\tag{4}
$$

**Theorem 1.** *The optimization problem defined in Equation (4) can be expressed as an equivalent linear programming (LP) problem.*

*Proof.* We simplify the definition of $v$ in the third constraint and rewrite the optimization problem as:

$$
\begin{aligned}
\min_{M,v} \quad & \mathrm{Tr}(MG) + \lambda v^\top R v \\
s.t. \quad & \langle M, E \rangle = B \\
& (M.e)_i \leq C_{max}, \forall i \\
& M_{ij} \leq v_i, \forall i,j \\
& v_i, M_{ij} \in \{0,1\}, \forall i,j
\end{aligned}
\tag{5}
$$

The constraint $M_{ij} \leq v_i, \forall i,j$ denotes that if row $i$ in $M$ has at least one entry as 1, then $v_i$ has to be 1. If row $i$ in $M$ has all entries as 0, then $v_i$ is free to be 0 or 1. However, we are solving a minimization problem with $v^\top R v$ in the objective, and $R$ has only non-negative entries; this criterion will force $v_i$ to be equal to 0, as that will result in a better (lower) value of the objective. This shows that the constraint $v_i = \min(1, (M.e)_i), \forall i$ in Equation (4) is equivalent to the linear constraint $M_{ij} \leq v_i, \forall i,j$ in Equation (5).

The first term in the objective function can be expressed as a linear term: $\mathrm{Tr}(MG) = \sum_{i,j} G_{ij}.M_{ji}$. Also, let $z_{ij} = v_i.v_j$. Clearly, $Z$ is a binary matrix of size $N \times N$ with all entries 0 or 1. The second term in the objective can then be written as:

$$
v^\top R v = \sum_{i,j} z_{ij}.r_{ij}
\tag{6}
$$

The optimization problem can thus be expressed as:

$$
\begin{aligned}
\min_{M,v,Z} \quad & \sum_{i,j} G_{ij}.M_{ji} + \lambda \sum_{i,j} z_{ij}.r_{ij} \\
s.t. \quad & \sum_{i,j} M_{ij} = B \\
& z_{ij} = v_i.v_j, \forall i,j \\
& (M.e)_i \leq C_{max}, \forall i \\
& M_{ij} \leq v_i, \forall i,j \\
& v_i, M_{ij}, Z_{ij} \in \{0,1\}, \forall i,j
\end{aligned}
\tag{7}
$$

Now, we attempt to express the quadratic equality $z_{ij} = v_i.v_j, \forall i, j$ as a linear term. The quadratic equality implies that $z_{ij}$ equals 1 only when both $v_i$ and $v_j$ are 1 and equals 0 otherwise. This can be expressed as the linear inequality $v_i + v_j \leq 1 + 2z_{ij}, \forall i, j$. From the inequality, we note that when both $v_i$ and $v_j$ are 1, $z_{ij}$ is forced to be 1. When $v_i$ and $v_j$ are both 0, or one of them is 0 and the other one is 1, $z_{ij}$ is free to be 0 or 1. Using the same argument as before, we note that we are solving a minimization problem with $\sum_{i,j} z_{ij}.r_{ij}$ in the objective and $R$ has only non-negative entries; thus, the nature of the problem will force $z_{ij}$ to be 0 as it will produce a lower value of the objective. Replacing the quadratic equality with the linear inequality, we express the optimization problem as follows:

$$
\begin{aligned}
\min_{M,v,Z} \quad & \sum_{i,j} G_{ij}.M_{ji} + \lambda \sum_{i,j} z_{ij}.r_{ij} \\
s.t. \quad & \sum_{i,j} M_{ij} = B \\
& v_i + v_j \leq 1 + 2z_{ij}, \forall i, j \\
& (M.e)_i \leq C_{max}, \forall i \\
& M_{ij} \leq v_i, \forall i, j \\
& v_i, M_{ij}, Z_{ij} \in \{0, 1\}, \forall i, j
\end{aligned}
\tag{8}
$$

In this optimization problem, both the objective function and the constraints are linear in the variables $M$, $v$ and $Z$. It is thus a linear programming (LP) problem.

$\square$

As mentioned in Section 3.2, we vectorize the variables $M$, $v$ and $Z$, append them one below the other and express the objective function and the constraints in terms of this new variable. The integer constraints are then relaxed into continuous constraints and the problem is solved using an off-the-shelf LP solver. After obtaining the continuous solution, we recover the integer solution of our variable of interest $M$, using a rounding approach where the $B$ highest entries in $M$ are reconstructed as 1 and the other entries as 0, observing the constraints.

## B STUDY OF QUERY BUDGET

The objective of this experiment was to study the effect of the query budget $B$ on the AL performance. The results on the Flickr dataset are shown in Figure 4 for query budgets 200, 300 and 400. As explained before, these budgets were imposed only for the region-level and binary-level annotation baselines (*RAL, RR, EE* and our method). The *Entropy* and *Coreset* methods still annotated the same number of images (48) at the pixel-level in each AL iteration, to represent an upper bound on the AL performance among the methods studied.

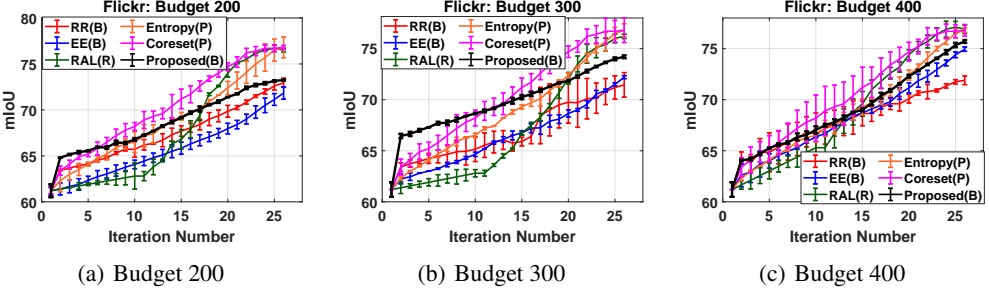

(a) Budget 200       (b) Budget 300       (c) Budget 400

Figure 4: Study of query budget on the Flickr dataset. Best viewed in color. The results with budget 400 are the same as those in Figure 2(a) and are included here for comparison.

| Budget | RR(B) | EE(B) | RAL(R) | Entropy(P) | Coreset(P) | Proposed(B) |
|--------|-------|-------|--------|-----------|-----------|-------------|
| **200** | $72.95 \pm 0.47$ | $71.83 \pm 0.66$ | $76.50 \pm 0.23$ | $76.8 \pm 0.58$ | $76.8 \pm 0.57$ | $73.25 \pm 0.09$ |
| **300** | $71.45 \pm 1.20$ | $72.20 \pm 0.14$ | $76.2 \pm 0.27$ | $76.8 \pm 0.58$ | $76.8 \pm 0.57$ | $74.20 \pm 0.17$ |
| **400** | $71.9 \pm 0.44$ | $74.95 \pm 0.21$ | $76.9 \pm 0.41$ | $76.8 \pm 0.58$ | $76.8 \pm 0.57$ | $75.7 \pm 0.13$ |

Table 5: Study of the query budget on the Flickr dataset. Final mIoU achieved by all the methods after 25 AL iterations (as shown in Figure 4) are depicted in the table. Here, **B** denotes binary-level annotation, **R** denotes region-level annotation and **P** denotes pixel-level annotation.

| Query Budget | Binary-Level | Region-Level | Pixel-Level |
|--------------|--------------|--------------|-------------|
| **200** | 2.78 | 133.33 | 156 |
| **300** | 4.16 | 200 | 156 |
| **400** | 5.56 | 266.67 | 156 |

Table 6: Approximate total time (**in hours**) to be expended for annotation (for the binary-level, region-level and pixel-level methods) over 25 AL iterations for different query budgets for the Flickr dataset. Query budget denotes the number of binary queries answered for binary-level annotation methods, and number of image regions annotated for the region-level annotation methods. Pixel-level annotation methods annotate all the $1,200$ unlabeled images at the pixel-level (48 images in each AL iteration, regardless of the budget).

Our framework consistently outperforms *RR* and *EE* (the two baselines for binary-level annotation) across all budgets, showing its usefulness across different query budgets. This result is particularly significant from a practical standpoint, where the available query budget is dependent on time, resources and other constraints of an application, and is different for different applications. Table 5 reports the final mIoU values for all the methods after 25 AL iterations for all the query budgets. From Figure 4 and Table 5, we note that the performance gap between our method and the pixel-level annotation baselines (*Entropy* and *Coreset*) increases with a reduction in query budget. This is intuitive, as with a reduction in query budget, the generalization capability of our model decreases while those obtained through *Entropy* and *Coreset* sampling remains the same (since their query budget remains the same). However, a reduction in query budget also means a reduction in total annotation time. Table 6 reports the total annotation time required by all the methods across the 25 AL iterations, for each query budget. We note that for query budgets 200, 300 and 400, our binary query framework results in 56.11 fold, 37.5 fold and 28.05 fold reduction in annotation time respectively, compared to the pixel-level annotations necessitated by the *Entropy* and *Coreset* baselines. From Table 6, we also note that region-level annotation can sometimes take more time than pixel-level annotation, depending on the number of regions annotated and the resolution of the images.

## C ABLATION STUDY

We conducted an ablation study to assess the importance of the uncertainty and redundancy terms in our formulation (Equation (4)). The results on the Flickr dataset (with query budget $400$) are depicted in Figure 5, which shows the performance of the proposed method, the proposed method without the redundancy term ($\lambda = 0$) and the proposed method without the uncertainty term ($\alpha = 0$). We note that, removing either the redundancy term or the uncertainty term adversely affects the performance of our method. This shows that it is important to consider both the class presence uncertainty (to query informative semantic classes) and the image redundancy (to avoid duplicate image queries) in our active learning framework for querying (image-class) pairs.

## D COMPUTATION TIME ANALYSIS

The goal of this experiment was to perform a comparative analysis of the computation time of the algorithms studied. The average time (in minutes) taken to query a batch of (image-class) pairs and retrain the deep model with the queried classes (one AL iteration), for the three datasets are reported in Table 7. The results are averaged over the 3 random runs and the 25 AL iterations for each run. The *Random-Random (RR)* method selects a batch of (image-class) pairs at random and there is no computation involved; it was hence excluded from this analysis. The algorithms were implemented

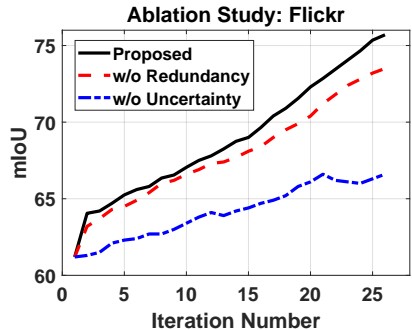

Figure 5: Ablation study results. Query budget $= 400$. Best viewed in color.

in Python on a Windows 10 Pro workstation with Intel(R) Xeon(R) Gold 5222 CPU @ 3.80GHz and 64GB RAM, equipped with Dual NVIDIA Quadro RTX 5000 GPUs with 16GB memory.

| Dataset | EE(B) | RAL(R) | Entropy(P) | Coreset(P) | Proposed(B) |
|---|---|---|---|---|---|
| **Flickr** | $17.10 \pm 0.86$ | $30.68 \pm 0.36$ | $13.77 \pm 0.28$ | $23.86 \pm 1.15$ | $25.32 \pm 0.62$ |
| **Cityscapes** | $19.51 \pm 0.40$ | $36.26 \pm 0.14$ | $16.9 \pm 1.04$ | $31.12 \pm 0.81$ | $27.94 \pm 0.57$ |
| **PASCAL** | $18.09 \pm 0.31$ | $28.4 \pm 0.76$ | $13.17 \pm 0.71$ | $22.61 \pm 1.92$ | $24.91 \pm 0.87$ |

Table 7: Average ($\pm$ std) time taken (in minutes) to query a batch of (image-class) pairs **and** retrain the deep model with the queried classes (one AL iteration), for all the methods, except *RR*. Here, **B** denotes binary-level annotation, **R** denotes region-level annotation and **P** denotes pixel-level annotation.

From Table 7, we note that the *Entropy* method has the least computation time. This is because the entropy of an image can be easily computed from the posterior probabilities furnished by the deep neural network. Further, training the deep network with binary-level annotations is much more difficult than training with pixel-level annotations (as illustrated in Section F.1); this explains the comparatively higher computation time of *EE* compared to *Entropy*. The proposed method has a higher computation time than *EE*, as it involves solving an LP minimization problem. The computation time of our framework is more or less similar to that of *Coreset*, which involves solving a mixed integer programming (MIP) problem (Sener & Savarese, 2018). The *RAL* method needs to compute the superpixels of all the unlabeled images. Also, it involves training the deep neural network with region-level annotations, which is a time-consuming process (similar to our binary-level annotations). It thus has a slightly higher computation time.

### D.1 IMPROVING THE COMPUTATION TIME OF THE PROPOSED FRAMEWORK

In this section, we present a couple of strategies, that can potentially improve the computational efficiency of our framework.

Computing the redundancy matrix $R$ (Equation(3)) involves quadratic complexity. We first note that $R$ needs to be computed only once in our framework (before the start of the AL iterations). Moreover, the theory of random projections can be used to reduce the computational overhead. Random projections have been successfully used to speed up computations, where an original data matrix $A \in \Re^{m \times D}$ is multiplied by a random projection matrix $X \in \Re^{D \times d}$ to obtain a projected matrix $B \in \Re^{m \times d}$ in the lower dimensional space $d$: $B = \frac{1}{\sqrt{d}}AX$, where $d \ll min(m, D)$ (Vempala, 2004). We plan to study this as part of our future research.

Further, solving the LP minimization problem (in Equation (8)) more efficiently can improve the computation time of our algorithm. Sridhar *et al.* Sridhar et al. (2013) proposed an algorithm to solve large-scale LP problems and showed that we can recover solutions of comparable quality by rounding an approximate LP solution instead of the exact one. These approximate LP solutions can be computed efficiently by applying a parallel stochastic-coordinate-descent method to a quadratic-

penalty formulation of the LP. For the sake of completeness, we discuss the main ideas here and request the interested reader to refer to Sridhar et al. (2013) for further details.

We first consider an LP minimization problem in its standard form:

$$
\begin{aligned}
\min_{x} \quad & c^\top x \\
s.t. \quad & Ax = b \\
& Cx \le d \\
& x \ge 0
\end{aligned}
\tag{9}
$$

Defining $y = d - Cx$, the inequality constraint reduces to $y \ge 0$. The variables $x$ and $y$ can then be concatenated into a single variable $z$ and the whole problem can be expressed in terms of this new variable. In our subsequent discussion therefore, we assume that we do not have any inequality constraint (similar to Sridhar et al. (2013)) and consider an LP minimization problem in the following form:

$$
\begin{aligned}
\min_{x} \quad & c^\top x \\
s.t. \quad & Ax = b \\
& x \ge 0
\end{aligned}
\tag{10}
$$

We next consider the following regularized quadratic penalty approximation to this LP, parameterized by a positive constant $\beta$, whose solution is denoted by $x(\beta)$:

$$
x(\beta) = \arg\min_{x \ge 0} f_\beta(x) = \quad c^\top x - \overline{u}^\top (Ax - b)
$$

$$
+ \frac{\beta}{2}\|Ax - b\|^2 + \frac{1}{2\beta}\|x - \overline{x}\|^2
\tag{11}
$$

where $\overline{u}$ and $\overline{x}$ are arbitrary vectors (can be set to zero). The stochastic coordinate descent (SCD) algorithm was used to solve Problem (11); the pseudocode is provided in Algorithm 2. In each step, the algorithm selects a component $i \in \{1, 2, \ldots n\}$ and takes a step in the $i^{th}$ component of $x$ along the partial gradient of Equation (11) with respect to this component, projecting if necessary to retain non-negativity. As evident from the pseudo-code, the procedure is very simple and the solution to the approximate LP can be computed efficiently. Please refer Sridhar et al. (2013) for convergence analysis results of this algorithm and the worst case complexity bounds for this approximate LP solver. A parallel version of the algorithm was also proposed, which is suitable for execution on multi-core, shared-memory architectures. In their empirical studies, the authors reported computational speedup by a factor of 2.8 to 9.0 (time taken by an off-the-shelf LP solver divided by the time taken by this method), with corresponding solution quality of 1.04 and 1.21 (ratio of the solution objective obtained by this method to that by an off-the-shelf LP solver) for solving LP minimization problems, similar to the one in this paper. Thus, this method has the potential to substantially reduce the computation time, without sacrificing too much on the solution quality. We plan to explore this framework to further improve the computation time of our algorithm, as part of future work.

---

**Algorithm 2** Stochastic Coordinate Descent Algorithm to solve Problem (11)

---

1: Select $x_0 \in \Re^n$
2: $j \leftarrow 0$
3: **loop**
4:     Select $i(j) \in \{1, 2, \ldots n\}$ randomly with equal probability
5:     Define $x_{j+1}$ from $x_j$ by setting
    $[x_{j+1}]_{i(j)} \leftarrow \max(0, [x_j]_{i(j)} - \frac{1}{L_{max}}[\nabla f_\beta(x_j)]_{i(j)})$, leaving other components unchanged
6:     $j \leftarrow j + 1$
7: **end loop**

---

# E    STUDY OF THE PARAMETER $C_{max}$

The goal of this experiment was to study the effect of the parameter $C_{max}$ (maximum number of semantic class queries that can be posed for each unlabeled image) in Equation (4), on the AL

performance. The results on the Flickr dataset for $C_{max} = \{4, 5, 7, 9\}$ and query budget 200 are depicted in Figure 6. A low value of $C_{max}$ ($\leq 4$) seems to adversely affect the learning performance. This is because, a low value restricts the number of classes that can be queried for each unlabeled image, and as a result, the algorithm may miss querying some of the informative classes. Otherwise, the performance of our framework is more or less robust to the choice of this parameter.

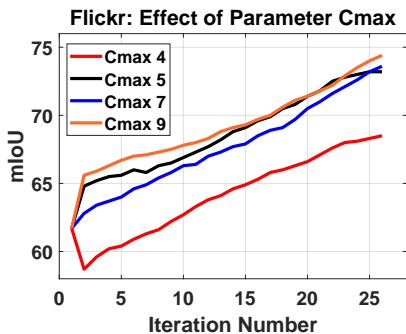

Figure 6: Study of the parameter $C_{max}$ on the Flickr dataset with query budget 200. Best viewed in color.

## F  IMPLEMENTATION DETAILS

We used the *DeepLabV3+* model with the ResNet101 backbone (pre-trained on ImageNet) as our base model due to its promising performance in image segmentation applications (Chen et al., 2018). It is constructed upon an encoder-decoder architecture where DeepLabv3 is used as an encoder or feature extractor which employs ResNet-101 as the backbone, together with atrous convolution layers to extract multi-scale contextual information; it also utilizes the Atrous Spatial Pyramid Pooling (ASPP) module for further fast processing and improved performance (Chen et al., 2018). In the ASPP module, 3 parallel $3 \times 3$ atrous convolutions with rates $(6, 12, 18)$ for output stride 16, and $(12, 24, 36)$ for output stride 8 were used in our experiments. Data augmentation was applied by randomly scaling the input images, and random left-right flipping during the training phase. The *adam* optimizer was used in our experiments. The details of the training parameters for all the datasets are provided in Table 8.

| Dataset | Learning Rate | Mini-batch Size | Momentum | Training Epochs |
|---|---|---|---|---|
| **Flickr** | 0.005 | 8 | 0.9 | 80 |
| **Cityscapes** | 0.0005 | 4 | 0.9 | 240 |
| **PASCAL VOC12** | 0.001 | 16 | 0.9 | 50 |

Table 8: Details of the training parameters for all the datasets.

### F.1  MODEL UPDATING WITH BINARY USER FEEDBACK

The DeepLabV3+ model takes the input images and their corresponding masks with the semantic class labels, into its training operation. For a given unlabeled image, the trained model similarly generates the segmentation mask and predicts the semantic classes present in it based on the generated mask. An example is shown in Figure 7. The model also maintains a vector of probabilities depicting the likelihood of the presence of each semantic class in the image. During the active learning iterations, as the user provides binary feedback regarding the presence / absence of a semantic class in a given image, the probability vector and the list of classes for that image are updated. There are four possible scenarios, as described below:

**(1) Model predicts a semantic class to be absent in an image, poses a binary query and receives a negative feedback from the oracle (class is absent)**: In this case, the predicted segmentation mask does not change; the probability vector is updated to reflect that the semantic class is not present in the image.

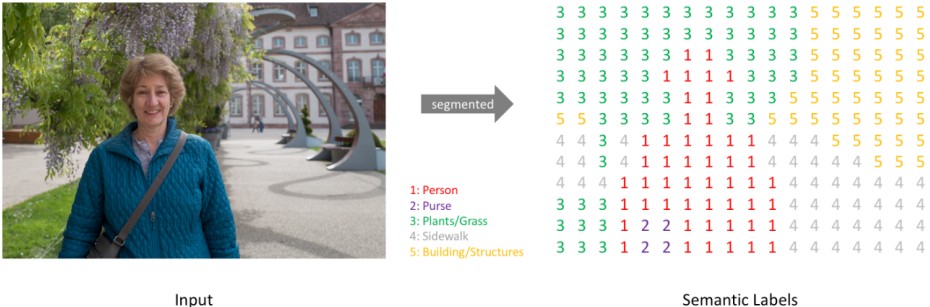

Figure 7: Image and model segmented output (pixel wise classification).

**(2) Model predicts a semantic class to be present in an image, poses a binary query and receives a positive feedback from the oracle (class is present)**: In this case also, the predicted segmentation mask does not change; the probability vector is updated to reflect that the semantic class is present in the image.

**(3) Model predicts a semantic class to be present in an image, poses a binary query and receives a negative feedback from the oracle (class is absent)**: In this case, the most probable predicted class (that is currently not included in the mask) will replace the class that is absent, in the segmentation mask; this class may get queried in a subsequent round.

**(4) Model predicts a semantic class to be absent in an image, poses a binary query and receives a positive feedback from the oracle (class is present)**: In this case, the new semantic class is accommodated in the segmentation mask as a "background" class, in an appropriate location. These locations are determined by searching the boundaries of the dominant class in the image, for pixels allocated to the background class; if no such positions are found, the corners of the image are searched. Our observation (through extensive experimental studies) reveals that the background class is usually located close to the dominant class or around the corners of the image.

Once the segmentation mask is updated, the model is retrained with this new information.

Note that, our binary query mechanism does not provide any information about the spatial locations of the semantic classes in an image. With reference to the above image, let us suppose that the model predicts the location of the "Person" class in the bottom right of the image, based on the current segmentation mask. While the image contains a person, the location is incorrectly predicted by the model. Even though the binary feedback mechanism does not provide any information about the spatial location of a semantic class in an image, it can still help in rectifying the location. For instance, it may so happen that a new semantic class is about to be introduced in the image as a "background" class and its most appropriate location is the bottom right of the image. In that case, the newly introduced class is accommodated there, and the position of the "Person" class is shifted to other available locations in the image with pixels allocated to the background class. Over time, as we receive more and more feedback about the presence / absence of different semantic classes in the image, the classes adjust themselves appropriately in the image; the predicted segmentation mask becomes more and more accurate, which increases the generalization capability of the model, and hence, the mIoU on the test set keeps increasing.

Our code will be made publicly available once our paper is accepted.

### F.2 VISUAL ILLUSTRATIONS

In the figures below, we have shown a few examples of how the segmentation mask of a particular test image evolves over the active learning (AL) iterations, as the deep model receives more and more binary feedback in our binary annotation framework. In Figure 8, we note the car in front (marked by the blue arrow) was absent in the $5^{th}$ AL iteration, but appears in the correct location in the $25^{th}$ AL iteration; we also note the portion of the hood of the car in the bottom left portion of

the image (marked by the red arrow) was not properly segmented in the $5^{th}$ AL iteration, but shows a much better segmentation in the $25^{th}$ iteration.

In Figure 9, the sidewalk in the bottom left portion of the image (marked by the red arrow) was not properly segmented in the $5^{th}$ AL iteration, but achieves the correct segmentation in the $25^{th}$ iteration, which increases the mIoU.

In Figure 10, the traffic light (shown by the yellow circle and marked by the red arrow) was absent in the $5^{th}$ AL iteration, but appears in the correct location in the $25^{th}$ AL iteration; also, the portion of the hood of the car in the bottom left portion of the image (marked by the blue arrow) was not properly segmented in the $5^{th}$ AL iteration, but shows a much better segmentation in the $25^{th}$ iteration.

These examples show that binary feedback can be effective in training a deep neural network for image segmentation.

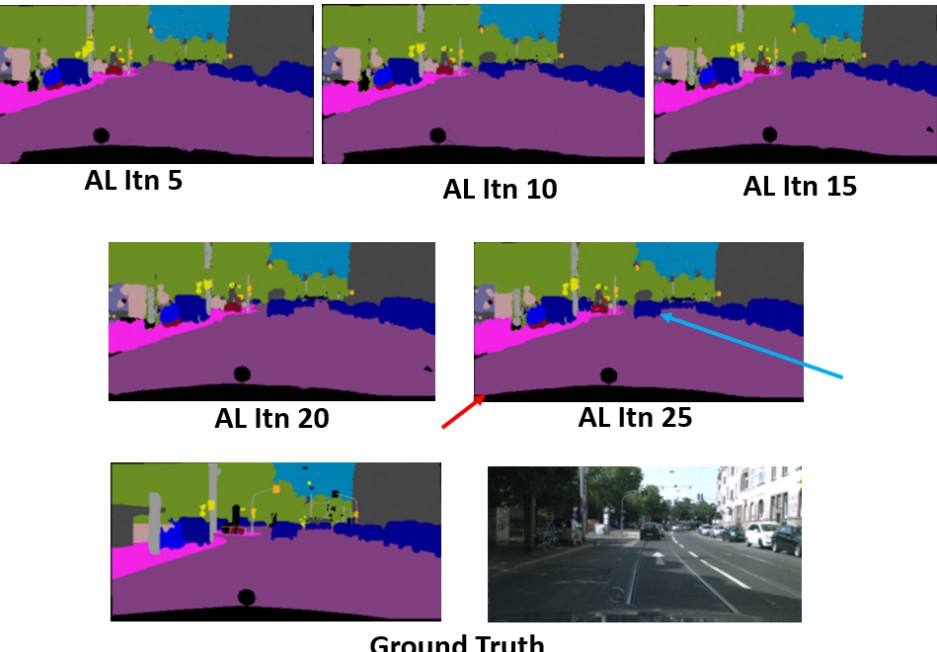

Figure 8: Visual Illustration 1. Best viewed in color.

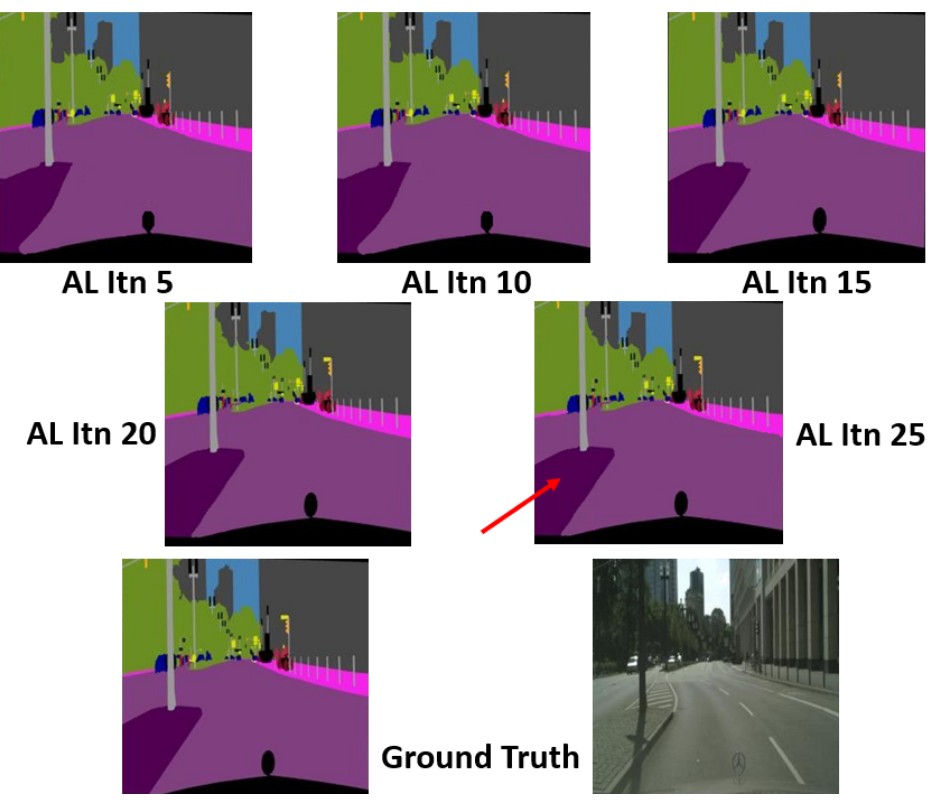

Figure 9: Visual Illustration 2. Best viewed in color.

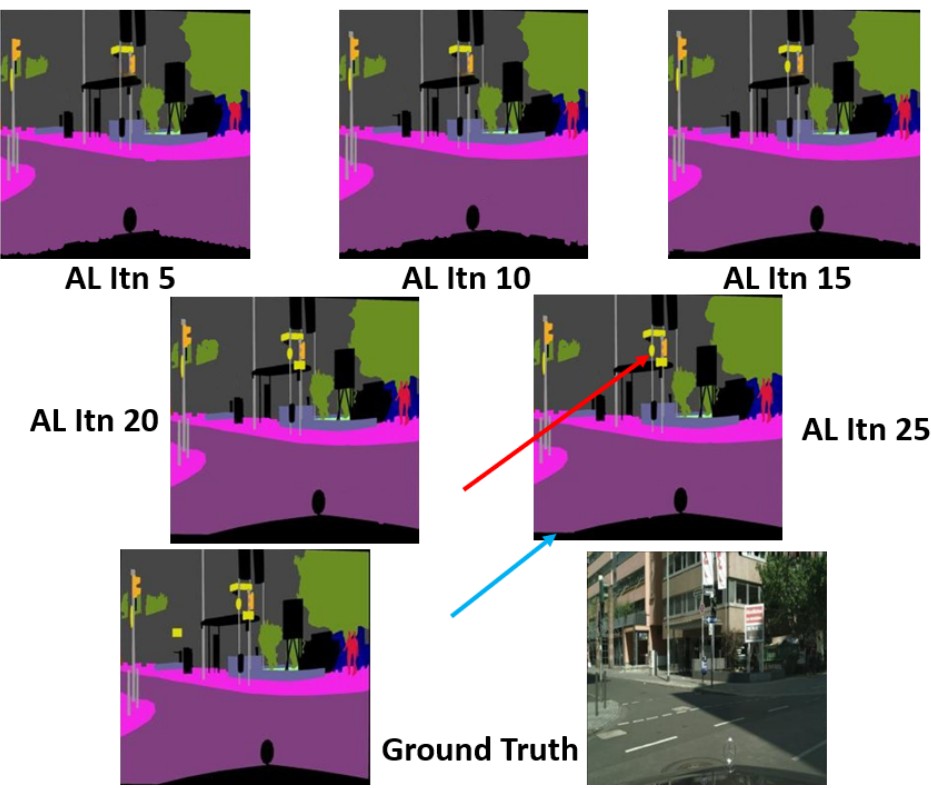

Figure 10: Visual Illustration 3. Best viewed in color.

## G   Study of Initial Training Set Size

In this experiment, we studied the effect of the size of the initial training set (with pixel-level annotations) on the AL performance. The results on the Cityscapes dataset, with training set sizes 300 and 500 are depicted in Figure 11. The results depict a very similar trend as Figure 2. The proposed method outperforms the other binary-level annotation techniques (*RR* and *EE*) and depicts competitive performance as the region-level (*RAL*) and pixel-level (*Coreset* and *Entropy*) annotation techniques. Thus, it can result in substantial savings in the human annotation effort in exchange of a marginal drop in accuracy. These results show the efficacy of the proposed method even when the size of the initial training set (containing pixel-level annotations) is small.

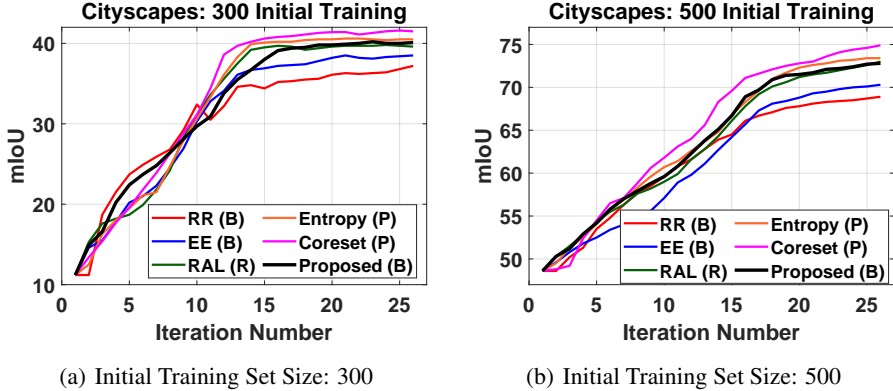

(a) Initial Training Set Size: 300    (b) Initial Training Set Size: 500

Figure 11: Study of initial set size on the Cityscapes dataset. The errorbars have been removed for better visualization. Best viewed in color.

## H   Comparison against Fully Supervised Baseline

In this experiment, we compared the performance of all the methods against a fully supervised baseline, where all the samples in the training and unlabeled sets were annotated at the pixel-level and the deep model was trained on the combined pool. The results are presented in Figure 12 where the fully supervised baseline is represented by the dashed flat line. After 25 AL iterations, our method achieves an mIoU that is very close to that obtained by the fully supervised baseline.

Table 9 reports the difference in mIoU between the fully supervised baseline and the proposed method after 25 AL iterations. We note that, just by using binary queries, our method achieves an mIoU that is very close to the fully supervised baseline. The table also shows the reduction in human annotation effort achieved by our method, compared to the fully supervised baseline. The fully supervised baseline requires all the images in the unlabeled set to be annotated at the pixel-level. From the table, it is evident that our binary query mechanism can result in substantial savings of human annotation effort. For the PASCAL dataset, for instance, our method achieves an mIoU that is only 0.74 units less than that achieved by the fully supervised baseline. However, our method also results in a 87.5 fold reduction in human annotation effort due to the binary queries, as opposed to the pixel-level queries for the fully supervised baseline. These results further corroborate the usefulness of our framework in drastically reducing human annotation effort, with a minor loss in performance, for image segmentation applications.

| Dataset | mIoU Difference | Annotation Effort Reduction |
|---|---|---|
| Flickr | 1.5 | 28.05 fold |
| Cityscapes | 1.87 | 134.89 fold |
| PASCAL VOC12 | 0.74 | 87.5 fold |

Table 9: Table showing the difference in mIoU between the fully supervised baseline and the proposed method after 25 AL iterations. The table also shows the reduction in annotation effort achieved by the proposed method after 25 AL iterations, compared to the fully supervised baseline.

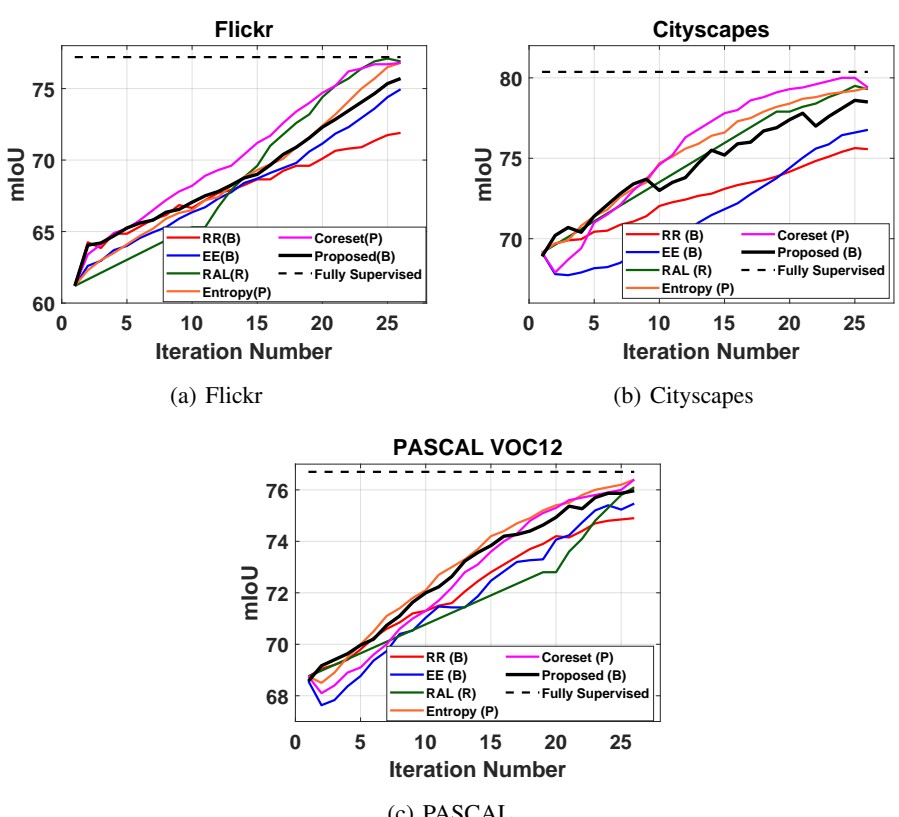

(a) Flickr

(b) Cityscapes

(c) PASCAL

Figure 12: Comparison against the fully supervised baseline. The errorbars have been removed for better visualization. Best viewed in color.

