# OpenReview forum: "Active Learning for Image Segmentation with Binary User Feedback"
_ICLR.cc/2024/Conference — Submitted to ICLR 2024_

### Official Review · Reviewer_eZUf · 2023-10-28

**Soundness:** 2 fair
**Presentation:** 3 good
**Contribution:** 2 fair
**Rating:** 5
**Confidence:** 3

**Summary:**

**This paper proposes an active learning algorithm that simplifies the task of data labeling in image segmentation. The proposed algorithm utilizes binary queries, asking about the existence of semantic classes in images, which appears to be a streamlined approach compared to conventional detailed annotation processes.** However, while the methodology is novel and the binary query concept intriguing, the paper might benefit from a more robust evaluation of its practical implications and limitations.

The empirical studies presented seem promising but might not fully encapsulate the algorithm's efficacy and applicability. Additionally, while the focus on reducing human effort is commendable, there appears to be **a critical assumption that binary queries are sufficient for image segmentation tasks, which might not always hold**. A crucial aspect to consider is the algorithm's reliance on an initial dataset. The effectiveness of the proposed method may be compromised if not furnished with a large number of images initially. This significant dependence on initial data volume underscores a vulnerability in its design, potentially restricting its practical applicability where acquiring ample relevant images is problematic. While the paper's innovative approach to reducing manual labeling efforts is noteworthy, its successful implementation seems contingent upon the availability of a robust initial image dataset. Such a requirement necessitates a thoughtful appraisal of its practical adaptability and overall efficacy in real-world scenarios.

Furthermore, the generalizability of this algorithm to other domains, such as object detection, is mentioned as a future direction, but it remains speculative without empirical validation. In conclusion, the paper presents an active learning for image segmentation but leaves room for a more comprehensive exploration of its practical potential and limitations.

**Strengths:**

This paper presents a new active learning algorithm for image segmentation in deep learning, boasting several notable strengths.

- The algorithm is characterized by a thoughtfully designed linear programming formulation. However, it's important to note that this design is primarily grounded in heuristics. This basis may influence the robustness and predictability of the algorithm, possibly affecting its overall efficacy and application in various contexts.

- The algorithm is user-friendly, making the annotation process considerably more manageable for humans. Simplifying the labeling task through binary queries regarding semantic classes in images fosters an environment where annotators can work more efficiently and effectively, reducing the complexity and burden of the annotation process.

- Performance-wise, the algorithm exhibits reasonable outcomes when furnished with an ample initial set of labeled images. With a sufficient starting dataset, the algorithm illustrates a notable level of effectiveness and utility, positioning itself as a potent tool for image segmentation tasks in deep learning. These observations affirm the viability and functionality of the proposed concept, indicating that the underlying idea holds merit and applicability.

**Weaknesses:**

This paper presents several notable areas for improvement and consideration.

- There is a concern regarding the Linear Programming (LP) formulation, as opposed to some aspects of my statements in strengths, which, while crucial, seems to necessitate significant memory, especially when handling large datasets with extensive semantic classes. This could potentially hinder the algorithm’s efficiency and applicability in broader contexts.

- The theoretical foundation of the proposed framework seems somewhat lacking. Since the formulations primarily rely on heuristics-based LP formulation, there’s an inherent uncertainty regarding the algorithm’s reliability and the specific conditions under which it might fail. A more robust theoretical backing would enhance the algorithm’s credibility and predictability.

- It’s essential to highlight the algorithm’s pronounced dependence on a considerable initial dataset for optimal functionality. This dependency could limit its utility in scenarios where access to such extensive initial data is restricted or impractical.

- The algorithm presented in this paper aims to reduce human labeling efforts, enhancing ease and efficiency in the data annotation process. However, it appears that this simplification comes at a cost, with the algorithm showing a certain level of compromised performance according to Figure 2 and Figure 3. The effort to make the labeling process more user-friendly and less labor-intensive seems to inadvertently lead to a trade-off, affecting the overall efficacy of the algorithm. This aspect suggests a delicate balance between facilitating human involvement and optimizing algorithm performance.

- The paper appears to omit consideration of recent advancements in active learning techniques. Notably, very few pixel annotation strategies, such as "PixelPick" [1], which emphasizes minimal pixel-based annotations in each image, and "BADGE" [2], "Balanced Entropy" [3], or "PowerBALD/PowerEntropy" [4], potentially offering a more effective approach than CoreSet, are overlooked. These examples, while not exhaustive, underscore the breadth of contemporary methodologies that could be instrumental in enriching the algorithm’s framework and applicability. By integrating these current developments, the paper could ensure a comprehensive alignment with the evolving landscape of the field, thereby bolstering the algorithm’s relevance and efficacy in the context of modern active learning paradigms.

[1] All you need are a few pixels: semantic segmentation with PIXELPICK, ICCVW 2021 - https://openaccess.thecvf.com/content/ICCV2021W/ILDAV/papers/Shin_All_You_Need_Are_a_Few_Pixels_Semantic_Segmentation_With_ICCVW_2021_paper.pdf

[2] Deep Batch Active Learning by Diverse, Uncertain Gradient Lower Bounds, ICLR 2020 - https://openreview.net/forum?id=ryghZJBKPS

[3] Active Learning in Bayesian Neural Networks with Balanced Entropy Learning Principle, ICLR 2023 - https://openreview.net/forum?id=ZTMuZ68B1g

[4] Stochastic Batch Acquisition: A Simple Baseline for Deep Active Learning, TMLR 2023 - https://openreview.net/forum?id=vcHwQyNBjW

**Minor Comment**

Page 4, Eq (1): The sign of entropy has been reversed. It should be $H_{ij}=-p_{ij}\log p_{ij} - (1-p_{ij})\\log (1-p_{ij})$. Otherwise, $H_{ij}$ would be a negative value.

**Questions:**

1. The proposed algorithm necessitates significant memory resources for the Linear Programming (LP) formulation. **Could the authors provide clarification regarding the memory consumption associated with the algorithm, particularly as the volume of unlabeled images increases?** Understanding how the algorithm's memory usage scales in response to larger datasets would be crucial for assessing its practical applicability and efficiency.

2. In evaluating Image Redundancy, i.e., Eq (3) on page 4, the authors have assigned a value of $0$ to negative cosine similarity values. However, it is worth considering whether this approach effectively captures the essence of redundancy. Negative cosine similarity values indicate an inverse redundancy, which suggests that setting these values to $0$ might not be the most informative choice. Using the absolute value of the cosine similarity could be a more reasonable alternative, as it would allow for a better understanding of redundancy relationships. **Could the authors please clarify the rationale behind assigning a $0$ value in this context and why it is deemed essential for accurately capturing redundancy in images?**

3. **Please elucidate the impact of the number of initially labeled images on the algorithm's performance.** This aspect is a pivotal assumption in this study, playing a crucial role in the algorithm’s effectiveness. It is essential to either validate this assumption or provide recommendations regarding the requisite quantity of initial images necessary for the algorithm to function optimally. Clarifying this matter will enhance the algorithm’s practical applicability and reliability in real-world scenarios.

---

> ### Author Response · Authors · 2023-11-17
> **Response to Reviewer eZUf: Part 1**
>
> Thank you very much for your valuable comments. We have revised the manuscript based on your feedback, and also provide responses here. All the revisions have been marked in red in the manuscript.
>
> **Q. There is a concern regarding the Linear Programming (LP) formulation, as opposed to some aspects of my statements in strengths, which, while crucial, seems to necessitate significant memory, especially when handling large datasets with extensive semantic classes. This could potentially hinder the algorithm’s efficiency and applicability in broader contexts.**
> We agree that larger datasets with extensive semantic classes will necessitate more memory. We propose two techniques to address this challenge:
>
> (1) Apply an uncertainty criterion to select a subset of images with the highest uncertainties. Then, apply the LP based selection only on the selected subset of images. Since the uncertainty of an image can be easily computed using entropy, this can substantially reduce the memory and computational overhead. Such a strategy has been used in previous AL research to deal with large datasets [1].
>
> (2) GPU-based parallel algorithms have been proposed to solve large-scale LP problems [2]. This can greatly reduce the computation time and is an effective way to solve large scale LP problems.
>
> Both of these will be investigated as part of future research.
>
> [1] S. Chakraborty, V. Balasubramanian, Q. Sun, S. Panchanathan, J. Ye. Active Batch Selection via Convex Relaxations with Guaranteed Solution Bounds. IEEE Transactions on Pattern Analysis and Machine Intelligence (TPAMI) 2015
>
> [2] J. Li, R. Lv, X. Hu, Z. Jiang. A GPU-Based Parallel Algorithm for Large Scale Linear Programming Problem. Intelligent Decision Technologies, 2011
>
> **We have mentioned this in the future work section of the revised paper.**
>
> **Q. The theoretical foundation of the proposed framework seems somewhat lacking. Since the formulations primarily rely on heuristics-based LP formulation, there’s an inherent uncertainty regarding the algorithm’s reliability and the specific conditions under which it might fail. A more robust theoretical backing would enhance the algorithm’s credibility and predictability.**
> We agree that the solution to the LP problem may sometimes fail (if the problem is infeasible / unbounded). However, in our wide range of experiments (across different datasets, query budgets, backbone network architectures and different initial training set sizes) we never encountered a situation where the solution to the LP problem failed.
>
> In case the solution to the LP problem fails in a particular AL iteration, a possible strategy is to use the class presence uncertainty matrix $H$ (in Equation 1) and query the $B$ image-class pairs with the highest class presence uncertainty; that is, for that AL iteration, we will be querying samples based on only the uncertainty criterion and not the redundancy criterion.
>
> Deriving the specific conditions under which the LP might fail is a topic of theoretical optimization research, and is outside the scope of this paper. Linear programming (LP) and quadratic programming (QP) solvers have been extensively used in AL research [3 ,4, 5].
>
> [3] Rita Chattopadhyay, Wei Fan, Ian Davidson, Sethuraman Panchanathan, and Jieping Ye. Joint transfer and batch-mode active learning. In International Conference on Machine Learning (ICML), 2013.
>
> [4] Zheng Wang and Jieping Ye. Querying discriminative and representative samples for batch mode active learning. In ACM SIGKDD International Conference on Knowledge Discovery and Data Mining, 2013.
>
> [5] Rita Chattopadhyay, Zheng Wang, Wei Fan, Ian Davidson, Sethuraman Panchanathan, and Jieping Ye. Batch Mode Active Sampling Based on Marginal Probability Distribution Matching. ACM Transactions on Knowledge Discovery from Data, 7(3), 2013.
>
>
> **Q. It’s essential to highlight the algorithm’s pronounced dependence on a considerable initial dataset for optimal functionality. This dependency could limit its utility in scenarios where access to such extensive initial data is restricted or impractical.**
> We agree this is a very valid concern. To address this, we have conducted two experiments on the Cityscapes dataset, where we reduced the size of the initial training set to **300** and **500** images with pixel-level annotations. **The results are presented in Figure 11 in Section G of the Appendix**. The results depict a very similar trend as Figure 2 in the paper. From the results we note that the mIoU furnished by our binary query based AL method is still very much competitive to the pixel-level and region-level annotation methods, and it outperforms the other binary level annotation methods. Thus, our method is still effective when the size of the initial training set is much smaller.

---

> ### Author Response · Authors · 2023-11-17
> **Response to Reviewer eZUf: Part 2**
>
> **Q. The algorithm presented in this paper aims to reduce human labeling efforts, enhancing ease and efficiency in the data annotation process. However, it appears that this simplification comes at a cost, with the algorithm showing a certain level of compromised performance according to Figure 2 and Figure 3. The effort to make the labeling process more user-friendly and less labor-intensive seems to inadvertently lead to a trade-off, affecting the overall efficacy of the algorithm. This aspect suggests a delicate balance between facilitating human involvement and optimizing algorithm performance.**
> Thank you for this insightful comment. As you have correctly pointed out, the main goal of this research is to study the trade-off between the human labeling effort and the overall efficiency of the algorithm. It is expected that, as we attempt to develop novel query mechanisms to reduce the labeling burden on the human oracles, we have to compromise on the performance of the algorithm. Since an AL algorithm with binary annotation did not exist, this trade-off could not be studied for image segmentation applications; to our knowledge, the proposed framework is the first of its kind to develop a binary query based annotation mechanism for AL in image segmentation, which enables us to study this trade-off. Our analysis revealed that we can often save substantial amounts of human annotation effort, in exchange for a minor loss in accuracy. For instance, from the results in Figure 3, we note that the final mIoU achieved by our binary query method is very close to that achieved by Coreset (less than 0.5% difference), which requires pixel-level annotations for the queried images; however, from Table 3, we note that the total annotation time required by Coreset is $134.89$ times greater than our method. We believe that the study of such a trade-off can provide valuable insights during the design and development of image segmentation applications.
>
> **Q. The paper appears to omit consideration of recent advancements in active learning techniques. Notably, very few pixel annotation strategies, such as “PixelPick” [1], which emphasizes minimal pixel-based annotations in each image, and “BADGE” [2], “Balanced Entropy” [3], or “PowerBALD/PowerEntropy” [4], potentially offering a more effective approach than CoreSet, are overlooked. These examples, while not exhaustive, underscore the breadth of contemporary methodologies that could be instrumental in enriching the algorithm’s framework and applicability. By integrating these current developments, the paper could ensure a comprehensive alignment with the evolving landscape of the field, thereby bolstering the algorithm’s relevance and efficacy in the context of modern active learning paradigms.**
> Thank you for this suggestion. We have added the mentioned references in the revised version of the paper. AL techniques like BADGE, BALD and Balanced Entropy require all the pixels to be annotated in the queried images. Since our annotation mechanism is fundamentally different (binary annotation), we did not perform an exhaustive comparison with state-of-the-art AL techniques. Rather, we selected our comparison baselines to cover a wide variety of annotation strategies: pixel-level annotation, region-level annotation and binary level annotation. Within the first category, we used Coreset (a state-of-the-art AL technique) and Entropy (a widely used AL technique) as our comparison baselines. Since our main contribution is a novel annotation mechanism for AL in image segmentation, we selected the comparison baselines to cover different types of annotation strategies, rather than only the state-of-the-art AL methods, which require pixel-level annotation.
>
> **Q. Entropy equation**
> Thank you very much for pointing this out. **We have corrected Equation (1) in the revised version of the paper.**
>
> **Q. Image Redundancy computation**
> We agree that there may be better ways to handle the situation where the cosine similarity produces a negative value. We assign a value of $0$ to entries where the cosine similarity is negative, so that the matrix $R$ does not contain any negative entries; this is critical in the proof of Theorem 1.
>
> An alternative idea may be to compute the redundancy matrix R and normalize all the values to be within a given range (say, $0$ to $1$) by applying an appropriate transformation. This will be investigated as part of our ongoing work.
>
> **Q. Generalizability to object detection**
> We agree with your concern and have removed this statement from the future work section.
>
> We hope these answer your questions. If you have any additional questions, we will be happy to answer.

---

> > ### Comment · Reviewer_eZUf · 2023-11-21
> >
> > I am thankful to the authors for taking the time to address my questions and concerns. In light of their responses, I maintain my initial score.

---

> > > ### Author Response · Authors · 2023-11-21
> > >
> > > Thank you very much for your time to read through our rebuttal.

---

### Official Review · Reviewer_ZnPV · 2023-10-29

**Soundness:** 3 good
**Presentation:** 2 fair
**Contribution:** 3 good
**Rating:** 6
**Confidence:** 4

**Summary:**

This paper proposes an active learning algorithm for image segmentation. The new method identifies a batch of informative images and a list of semantic classes for each image by a constrained optimization problem. The human annotator merely needs to answer whether a given semantic class is present or absent in a given image. The experimental results show that the proposed method consumes less time than pixel-level annotations and region-level annotations, and its annotation results are better than other comparative binary-level annotations.

**Strengths:**

This paper applies an active learning framework to image segmentation, which poses only binary (yes/no) queries to the users.

**Weaknesses:**

(1) Algorithm 1 does not include all the details of the method.

(2) The article focuses on how to sample images and provide object classes through optimization models, and there is less and vague introduction on how to iteratively achieve pixel level annotation of images.

**Questions:**

(1)	 Algorithm 1 does not include all the details of the method, that is, Algorithm 1 only introduces the method of selecting the unlabeled images and the corresponding semantic classes, but does not mention how the annotator answers yes/no, nor does it mention how to achieve higher precision pixel level annotation iteratively after obtaining M.

(2)	How to annotate an image at the pixel level only by determining the classes of the objects in the image? In other words, if all the classes of the objects in the image are correct, how to accurately locate these objects and achieve pixel level annotation?

---

> ### Author Response · Authors · 2023-11-17
> **Response to Reviewer ZnPV**
>
> Thank you very much for your valuable comments.
>
> **Q1. Algorithm 1 does not include all the details of the method, that is, Algorithm 1 only introduces the method of selecting the unlabeled images and the corresponding semantic classes, but does not mention how the annotator answers yes/no, nor does it mention how to achieve higher precision pixel level annotation iteratively after obtaining M**
>
> **Q2. How to annotate an image at the pixel level only by determining the classes of the objects in the image? In other words, if all the classes of the objects in the image are correct, how to accurately locate these objects and achieve pixel level annotation?**
>
> For both these questions, we would request the reviewer to kindly refer to Sections F.1 and F.2 in the Appendix, where we have explained in detail how we locate the objects in a given image and achieve pixel level predictions with binary feedback. We also have a couple of visual illustrations in Section F.2, where we have demonstrated how an object gradually shifts to its correct location in a particular test image, as the model receives more and more binary feedback through the AL iterations.
>
> We hope these answer your questions. If you have any additional questions, we will be happy to answer.

---

> > ### Comment · Reviewer_ZnPV · 2023-11-21
> >
> > Thank you for the authors‘ explanation. The complete algorithm process should be reflected in the main text. This article lacks a certain theoretical foundation and analysis. Therefore, I maintain my previous opinion.

---

### Official Review · Reviewer_cBzD · 2023-10-31

**Soundness:** 3 good
**Presentation:** 3 good
**Contribution:** 3 good
**Rating:** 5
**Confidence:** 5

**Summary:**

In this paper, the authors proposed an active learning algorithm for image segmentation, which queries binary feedback to ascertain the presence or absence of a semantic class within an unlabeled image. The authors identified the informative image-class pairs by considering both the class presence uncertainty and image redundancy. Furthermore, they conduct experiments and user studies on three image segmentation benchmarks to assess the efficacy of the proposed method.

**Strengths:**

Compared to pixel-wise or region-wise annotations, binary (yes/no) queries are more time-efficient and less labor-intensive.

**Weaknesses:**

Weakness
1.Both metrics of uncertainty and diversity are lack of novelty. Besides, in computing class presence uncertainty, the calculation of the probability that image i contains the semantic class j remains unclear. Is it the average probability of pixels belonging to the semantic class j within image i?
2.The rationale of selecting informative image-class pairs through an optimization problem, rather than designing a sampling strategy, lacks clarity.
3.The authors employed 1,500 images with pixel-wise annotations to construct the initial training set. Despite the efficiency gains from binary queries in subsequent AL rounds, the annotation cost for the initial training set remains substantial. Moreover, I have reservations about the impact of the initial training set size on AL performance. In other words, if there are fewer pixel-wise annotated samples in the initial training set, will the binary-query-based AL still yield effective results?
4.The authors did not investigate recent advancements in active learning (AL), and the related work section should be updated accordingly. Besides, the authors did not compare with state-of-the-art AL methods.

**Questions:**

see above.

---

> ### Author Response · Authors · 2023-11-17
> **Response to Reviewer cBzD**
>
> Thank you very much for your valuable comments. We have revised the manuscript based on your feedback, and also provide responses here. All the revisions have been marked in red in the manuscript.
>
> **Q. Both metrics of uncertainty and diversity are lack of novelty. Besides, in computing class presence uncertainty, the calculation of the probability that image i contains the semantic class j remains unclear. Is it the average probability of pixels belonging to the semantic class j within image i?**
> We agree that the uncertainty and diversity metrics have been explored in AL research before and are not novel. Our primary novelty in this research is the formulation of an AL technique for image segmentation which poses only binary queries to the human annotators. To the best of our knowledge, such a query technique has not been explored before in the context of image segmentation. Since our main contribution is not the development of an active sampling criterion, rather it is the development of a novel query strategy, we used an existing criterion based on uncertainty and diversity in our work.
>
> You are absolutely correct. The probability that image i contains the semantic class $j$ is the average probability of pixels belonging to the semantic class $j$ within image $i$. **We have clarified this in the revised version of our paper, just before Equation (1)**.
>
> **Q. The rationale of selecting informative image-class pairs through an optimization problem, rather than designing a sampling strategy, lacks clarity.**
> Our objective was to select a batch of image-class pairs that are both informative individually, and mutually diverse (non-redundant). It may be a little challenging to design a sampling strategy to simultaneously optimize two criteria. A possible sampling strategy may be to first select $K$ (where $K > B$, the actual batch size) image-class pairs based on the informativeness criterion only, and then select the $B$ most diverse image-class pairs from the selected subset. This would be a two-step solution process.
>
> In contrast, in our algorithm, we devise two separate terms for the informativeness and redundancy criteria and formulate an optimization problem to optimize them simultaneously. Solving the optimization problem directly gives us the $B$ image-class pairs that are both informative and non-redundant, in a single step.
>
> **Q. The authors employed 1,500 images with pixel-wise annotations to construct the initial training set. Despite the efficiency gains from binary queries in subsequent AL rounds, the annotation cost for the initial training set remains substantial. Moreover, I have reservations about the impact of the initial training set size on AL performance. In other words, if there are fewer pixel-wise annotated samples in the initial training set, will the binary-query-based AL still yield effective results?**
> We agree that this is a very valid concern. We have conducted two experiments on the Cityscapes dataset, where we reduced the size of the initial training set to **300** and **500** images with pixel-level annotations. **The results are presented in Figure 11 in Section G of the Appendix.** The results depict a very similar trend as Figure 2 in the paper. From the results we note that the mIoU furnished by our binary query based AL method is still very much competitive to the pixel-level and region-level annotation methods, and it outperforms the other binary level annotation methods. Thus, our method is still effective when the size of the initial training set is much smaller.
>
> **Q. The authors did not investigate recent advancements in active learning (AL), and the related work section should be updated accordingly. Besides, the authors did not compare with state-of-the-art AL methods.**
> We have updated the related work section to include a few recent references on active learning.
>
> State-of-the-art AL methods require the human annotators to annotate all the pixels in the queried images. Since our annotation mechanism is fundamentally different (binary annotation), we did not perform an exhaustive comparison with state-of-the-art AL techniques. Rather, we selected our comparison baselines to cover a wide variety of annotation strategies: pixel-level annotation, region-level annotation and binary level annotation. Within the first category, we used Coreset (a state-of-the-art AL technique) and Entropy (a widely used AL technique) as our comparison baselines. Since our main contribution is a novel annotation mechanism for AL in image segmentation, we selected the comparison baselines to cover different types of annotation strategies, rather than only the state-of-the-art AL methods, which require pixel-level annotation.
>
> We hope these answer your questions. If you have any additional questions, we will be happy to answer.

---

> > ### Author Response · Authors · 2023-11-22
> > **Requesting Reviewer cBzD to kindly check our response**
> >
> > Dear Reviewer cBzD,
> >
> > Thank you for your suggestions. We have updated the paper based on your feedback and have also conducted additional experiments to demonstrate the usefulness of our method when the size of the initial training set is small. We would request you to kindly check our response and the revised version of the paper and let us know if you have any further comments.
> >
> > Best regards,
> >  - Authors of Paper 752.

---

> ### Author Response · Authors · 2023-11-23
> **Requesting Reviewer cBzD to kindly check our response**
>
> Dear Reviewer cBzD,
>
> We wanted to check back with you to see if you got a chance to review our rebuttal and the revised version of the paper, and whether you had any further questions. If not, we would like to request you to kindly consider increasing your score.
>
> Best regards,
>  - Authors of Paper 752.

---

### Official Review · Reviewer_nZpY · 2023-11-08

**Soundness:** 2 fair
**Presentation:** 2 fair
**Contribution:** 3 good
**Rating:** 5
**Confidence:** 3

**Summary:**

The paper proposes to use Active Learning (AL) to query for weak labels for the task of semantic segmentation. The proposed approach is evaluated on three different datasets with two different backbones for the chosen architecture. Additionally a small user study is performed to motivate the benefits provided by the proposed approach.

**Strengths:**

Interesting idea to combine existing AL techniques with weak labels.

Very helpful to motivate the proposed approach with a user study and actual labelling effort measured in annotation time.

The proposed approach is evaluated on multiple datasets of varying difficulty/scope, additionally two different backbones are used in the experiments.

**Weaknesses:**

# General comments
Overall the idea is interesting, while incremental in novelty. The presentation and language of the submission have room for improvement. As the paper touches weak-labelling and active learning, it could be better embedded especially in the literature/related work on semi-supervised learning, self-supervised learning and especially learning from weak labels.

Minor hints on language, there are more of those, i suggest to consider an additional proof reading step by a native speaker:
* "to induce a neural network" not sure what this formulation refers to, it is used at multiple places.
* "which entails much lesser annotation effort" consider rephrasing
* "furnishing the highest prediction entropy" not sure 'furnishing' is an optimal choice of wording here

# Related work
While (Settles, 2010) is certainly an important reference, however, there are more current survey papers that can be cited here, e.g. https://arxiv.org/pdf/2203.13450.pdf

"Active learning for image segmentation has been comparatively less explored than other applications."
I disagree with that statement, with semantic segmentation being algorithmically very close to classification, I'd argue semantic segmentation is one of the prime applications of AL methods. See e.g. the references the authors themselves use, (Casanova et al., 2020), (Kasarla et al., 2019), (Mackowiak et al., 2018), (Vezhnevets et al., 2012), (Golestaneh & Kitani, 2020) and more, like

Siddiqui, Yawar, Julien Valentin, and Matthias Nießner. "Viewal: Active learning with viewpoint entropy for semantic segmentation." Proceedings of the IEEE/CVF conference on computer vision and pattern recognition. 2020.

Xie, Shuai, et al. "Deal: Difficulty-aware active learning for semantic segmentation." Proceedings of the Asian conference on computer vision. 2020.

Learning from weak labels seems to be missing from the comparisons, i suggest adding a discussion or better experimental comparison, some suggestions below.

Olmin, Amanda, et al. "Active Learning with Weak Labels for Gaussian Processes." arXiv preprint arXiv:2204.08335 (2022).

Wu, Jian, et al. "Weak-labeled active learning with conditional label dependence for multilabel image classification." IEEE Transactions on Multimedia 19.6 (2017): 1156-1169.

Younesian, Taraneh, et al. "Qactor: Active learning on noisy labels." Asian Conference on Machine Learning. PMLR, 2021.

Wu, Jian, et al. "Weak-labeled active learning with conditional label dependence for multilabel image classification." IEEE Transactions on Multimedia 19.6 (2017): 1156-1169.

Lu, Zhiwu, et al. "Learning from weak and noisy labels for semantic segmentation." IEEE transactions on pattern analysis and machine intelligence 39.3 (2016): 486-500.

# Implementation Details

## User Study
Very valuable to provide some evidence on the time required for manual labelling.
Unfortunately the user study has been conducted with a relatively low number of Annotators (3) and images (10) and only the mean was reported, potentially averaging over very extreme differences in annotator performance.

The reported numbers are still valuable, but the study could be improved. To understand the results better it would help to elaborate on the proficiency of the annotators (un-trained?) and give a reference to the annotation tool used. Different tools can have very varying suitability for the three different tasks that were compared.

With the low number of images and annotators exhaustive statistics seem non-appropriate, but providing some guidance on the spread of the values, e.g., standard-deviation (across all dimensions or across images), would help in interpreting them.

## Experimental setup
For a study as the presented one the split into train, test and pool is actually very important, however details on how the split was realized are lacking. Additionally there is no validation set mentioned, usually i'd assume the validation was used to tune the training of the chosen architecture on the chosen dataset.

Cityscapes provides 20k weakly annotated frames, given, that the authors propose a AL + weak label scheme, those would be a prime candidate for more exhaustive experiments.

Side-note: i find the graphs a bit hard to read due to the chosen style of presentation, different line styles and markers, as well as larger images could help.

**Questions:**

* Which labelling tool was used for the user study?
* Were the annotators trained at the task or did they do labelling for the first time?
* How was the train, test, pool split realized? Same strategy for the different datasets?
* Why is the test set rather small?
* Was there no validation set?
* How were the parameters for the training of the DNN chosen?
* The reported numbers do not seem to match the full size of e.g., Cityscapes, any reason to not using the full dataset for the experiments?
* How do the results compare to a random baseline?
* How do the results compare to a baseline (same architecture) trained on the full set (train + pool) evaluated on the custom test split?

---

> ### Author Response · Authors · 2023-11-17
> **Response to Reviewer nZpY**
>
> Thank you very much for your comments. We have revised the manuscript based on your feedback, and also provide responses here. All the revisions have been marked in red in the manuscript.
>
> # General Comments
>
> **Q. “to induce a neural network” not sure what this formulation refers to, it is used at multiple places.**
> We have updated this statement and have replaced “induce” with “train”.
>
> **Q. “which entails much lesser annotation effort” consider rephrasing**
> We have updated this statement and have replaced “entails” with “requires”.
>
> **Q. “furnishing the highest prediction entropy” not sure 'furnishing' is an optimal choice of wording here**
> We have updated this statement and have replaced “furnishing” with “producing”.
>
>
>
> # Related Work
>
> We have included all the mentioned references in the revised version of the paper. We have removed the statement: "Active learning for image segmentation has been comparatively less explored than other applications."
>
>
>
>
> # Implementation Details
>
> **Q. Annotation tool used**
> We used the LabelMe annotation tool for the user study. We have mentioned this in Section 4.5 of the revised version of the paper and have also included a reference.
>
> **Q. Proficiency of the annotators**
> The annotators did not have any training specifically for the annotation task. This was done to mimic a real-world scenario where the annotators may not have specific knowledge / training about the application in question.
>
> **Q. Standard deviation of the values**
> We have included the standard deviation values in Table 1 of the revised version.
>
> **Q. How was the train, test, pool split realized? Same strategy for the different datasets?**
> Following the convention in AL research, a given dataset was split at random to generate the training, unlabeled and test sets. We repeated the experiment for three different random splits and averaged the results, to rule out any effects of randomness.
>
> **Q. Was there no validation set?**
> We did not use any separate validation set. This is because, the main premise of AL is that we do not have a large amount of labeled data; using a separate validation set (besides the training set) will be contradictory to the conventional AL setup. During training the model, we randomly used 10% of the training set itself as the validation set.
>
> **Q. Cityscapes provides 20k weakly annotated frames, given, that the authors propose a AL + weak label scheme, those would be a prime candidate for more exhaustive experiments.**
> We are aware that the Cityscapes dataset provides several frames that are weakly annotated. However, the annotation scheme that we proposed in this paper is binary; that is, annotators are asked whether a particular semantic class is present in a particular image, and they simply have to answer YES / NO. This form of annotation is much weaker than the weakly annotated frames.
>
> **Q. How were the parameters for the training of the DNN chosen?**
> We followed the training parameter settings in the following papers in our work:
>
> [1] L. Chen, G. Papandreou, F. Schroff, H. Adam. Rethinking Atrous Convolution for Semantic Image Segmentation. arXiv:1706.05587v3, 2017
>
> [2] B. Kim, H. Choi, H. Jang, S. Kim. Resolution-Aware Design of Atrous Rates for Semantic Segmentation Networks. arXiv:2307.14179, 2023
>
> [3] F. Lin, T. Wu, S. Wu, S. Tian, G. Guo. Feature Selective Transformer for Semantic Image Segmentation. arXiv:2203.14124v3, 2022
>
> **Q. Why is the test set rather small? The reported numbers do not seem to match the full size of e.g., Cityscapes, any reason to not using the full dataset for the experiments?**
> No particular reason. We plan to conduct experiments on the full dataset as part of our ongoing research.
>
> **Q. How do the results compare to a random baseline?**
> Random sampling was included as a comparison baseline in all our experiments; the method RR denotes the baseline where the images, as well as the semantic classes were selected at random for active query.
>
> **Q. How do the results compare to a baseline (same architecture) trained on the full set (train + pool) evaluated on the custom test split?**
> We conducted an experiment to study the performance of our method compared to a fully supervised baseline (trained on the training set + the complete unlabeled pool). **The results are presented in Figure 12 and Table 9 in Section H of the Appendix.** These results further corroborate the usefulness of our framework in drastically reducing human annotation effort, with a minor loss in performance, for image segmentation applications. Please refer to section H of the Appendix for more details.
>
> We hope these answer your questions. If you have any additional questions, we will be happy to answer.

---

> > ### Author Response · Authors · 2023-11-22
> > **Requesting Reviewer nZpY to kindly check our response**
> >
> > Dear Reviewer nZpY,
> >
> > Thank you for your suggestions. We have updated the paper and have also conducted additional experiments based on your feedback. We would request you to kindly check our response and the revised version of the paper and let us know if you have any further comments.
> >
> > Best regards,
> >  - Authors of Paper 752.

---

> ### Author Response · Authors · 2023-11-23
> **Requesting Reviewer nZpY to kindly check our response**
>
> Dear Reviewer nZpY,
>
> We wanted to check back with you to see if you got a chance to review our rebuttal and the revised version of the paper, and whether you had any further questions. If not, we would like to request you to kindly consider increasing your score.
>
> Best regards,
>  - Authors of Paper 752.

---

### Author Response · Authors · 2023-11-20
**Request to all reviewers to reply to our rebuttal**

Dear Reviewers,

Thank you for your constructive comments and suggestions. Our point-by-point responses can be found below. We have also **added more experiments and revised our paper** based on your suggestions and comments. Since the end of the Rebuttal/Discussion phase is approaching, we hope that you can reply to our rebuttal and consider updating your scores, if we have addressed your concerns. Also, please let us know if there are any additional concerns or feedback.

Thank you very much.

Best regards,
 - Authors of Paper 752

---

### Author Response · Authors · 2023-11-21
**Request to reviewers to respond to our rebuttal**

Dear Reviewers nZpY, cBzD and ZnPV,

We wanted to check back with you to see if our rebuttal has addressed all your concerns and whether you had any further comments.

We sincerely appreciate all your comments, which have helped us improve the presentation of the paper significantly; **the additional experiments that we have conducted based on your suggestions have further demonstrated the usefulness of the proposed method.** Technically, our method is the first research effort to develop a binary query mechanism for active learning for image segmentation applications. Our results (and the user study conducted with human annotators) have demonstrated that the proposed technique can result in substantial savings of the human annotation effort, in exchange for a minor loss in accuracy. We believe that the proposed technique can be a promising candidate to save human annotation effort in real-world image segmentation applications.

Thank you very much.

Best regards,
 - Authors of Paper 752

---

### Meta-Review · Area_Chair_rwbg · 2023-12-05

**Metareview:**

This paper presents an active learning algorithm for image segmentation, where the labeler is only required to provide binary feedback (whether a class is present).

After the rebuttal and AC-reviewer discussion stage, the final scores of this paper are 5/5/5/6. One reviewer changed his/her score from 3 to 5 but it is still negative. The only positive reviewer (rating 6) did not show up in the discussion, while the other three reviewers arrived at a consensus of rejection. The authors sent some private comments to AC, which were well received and read by the AC, but they were not strong enough to overturn the reviewers' comments.

**Justification For Why Not Higher Score:**

The reviewers arrived at a consensus of rejection.

**Justification For Why Not Lower Score:**

N/A

---

### Decision · Program_Chairs · 2024-01-16

Reject